

# Classification of Arctic multilayer clouds using radiosonde and radar data

Maiken Vassel[1], Luisa Ickes[2], Marion Maturilli[3], and Corinna Hoose[1]

[1]Institute of Meteorology and Climate Research, Karlsruhe Institute of Technology, Karlsruhe, Germany
[2]Department of Meteorology, Stockholm University, Stockholm, Sweden
[3]Alfred Wegener Institute for Polar and Marine Research, Potsdam, Germany

**Correspondence:** Maiken Vassel (maiken.vassel@alumni.kit.edu), Luisa Ickes (luisa.ickes@misu.su.se)

**Abstract.** Multilayer clouds (MLC) occur more often in the Arctic than globally. In this study we present the results of a detection algorithm applied to radiosondes and radar from an one-year time period in Ny-Ålesund, Svalbard. Multilayer cloud occurrence was found on 29 % of the investigated days. These multilayer cloud cases are further analysed regarding the possibility of ice crystal seeding, meaning that an ice crystal can survive sublimation in a subsaturated layer between two cloud

layers when falling through this layer. For this we analyse profiles of relative humidity with respect to ice to identify super- and subsaturated air layers. Then the sublimation of an ice crystal of an assumed initial size of $r = 100$ μm on its way through the subsaturated layer is calculated. If the ice crystal still exists when reaching a lower supersaturated layer, ice crystal seeding can potentially take place. Seeding cases are found often, in 23 % of the investigated days. The identification of seeding cases is limited by the radar signal inside the subsaturated layer. Clearly separated multilayer clouds, defined by a clear interstice

in the radar image, do not interact through seeding (9 % of the investigated days). Since there are various deviations between the relative humidity profiles and the radar images, for the non-seeding cases an evaluation by manual visual inspection is additionally done.

## 1 Introduction

Clouds radiate downwards in the long-wave part of the spectrum and thereby warm the surface in the Arctic during most of

the year (Shupe and Intrieri, 2004). However, the correct representation of cloud fraction, cloud water content and its phase, particle size and cloud temperature is difficult but essential to improved weather forecasting (Barrett et al., 2017a, b). Therefore clouds are still a major contributor to uncertainty in both weather and climate prediction.

    In the recent years, an emphasis of research has been on Arctic mixed-phase clouds (Andronache, 2018; Morrison et al., 2012; Loewe et al., 2017). These clouds occur frequently in the Arctic, at all heights up to 6.5 km, and exist in the temper-

ature range between $-34\,°C$ to $0\,°C$ (Intrieri et al., 2002). They often consist of a supercooled liquid layer at cloud top and precipitating ice particles below. From the measurement point of view, Verlinde et al. (2007, 2013) described multilayered clouds as layers of variable lidar (light detection and ranging) signals inside a more or less continuous radar (radio detection and ranging) signal. With that they refer to multiple distinct liquid layers within one vertical extensive cloud. In contrast to multilayered clouds, multilayer clouds (MLCs) are described as two separate clouds with a clear visible interstice in between



(Tsay and Jayaweera, 1984; Intrieri et al., 2002; Khvorostyanov et al., 2001; Fleishauer et al., 2002; Liu et al., 2012). The coexistence of two clouds in different heights, in the Arctic often a boundary layer cloud and a higher mixed-phase cloud, can be explained by horizontally inhomogeneous advection (Luo et al., 2008). When large-scale meridional transport brings warm moist air into the Arctic, temperature and humidity inversions occur frequently (Nygård et al., 2014). Reaching supersaturation

and in the presence of sufficient cloud condensation nuclei (CCN) and ice nuclei (IN), this horizontal advection can result in cloud formation at multiple heights (Curry and Herman, 1985).

Christensen et al. (2013) analysed radar and lidar data collected by the satellites CloudSat (millimetre wavelength cloud pro-filing radar) and CALIPSO (Cloud–Aerosol Lidar and Infrared Pathfinder Satellite Observations) to investigate the occurence of MLCs. They found, excluding the Arctic, the global average occurrence of MLCs to be 11 % of the data. For the Arctic,

Liu et al. (2012) analysed similar satellite data of CloudSat and CALIPSO and found Arctic MLCs to occur between 17-25 % of the investigated time. The contribution of the MLCs to the seasonal variation of Arctic cloud coverage is only very weak. Cloud detection by satellites is challenging in the Arctic. A poor thermal and visible contrast between clouds and the underly-ing surface of snow and ice and small radiative fluxes from the cold polar atmosphere are only some of the uncertainties (Liu et al., 2012). Therefore and since the minimum considered layer thickness for separation was 960 m, Liu et al. (2012) assumed

their estimated MLC occurence most likely to be underestimated.

Microphysical interaction between MLC layers can happen through the seeder-feeder mechanism (Fleishauer et al., 2002; Avramov and Harrington, 2010; Hobbs and Rangno, 1998). This means that falling ice crystals from the upper cloud influence the lower cloud. However, ice formation in Arctic boundary layer clouds is not fully understood (Fridlind et al., 2012; Paukert and Hoose, 2014) and the frequency of seeding ice crystals from above into the lower cloud still needs to be investigated.

The objective of this study is to answer how often MLCs occur in the Arctic. Thereby we include an estimate for the possibility of the seeder-feeder mechanism between MLCs. For answering this question we present a MLC classification based on ground-based and in-situ measurements. In this study the first step is the analysis of radiosonde profiles to estimate the presence of MLCs. Radiosondes have the advantage to be relatively easy accessible in the Arctic. In this way the algorithm for MLC detection could easily also be applied to various other Arctic locations. However, the use of only radiosondes has

limitations and needs to be verified. For this we chose Ny-Ålesund, Svalbard, as an example study site where also radar data is available.

In Section 2 we present the datasets of radiosondes and radar used for the classification, we explain the methodology of the classification, and we consider the possibility of the seeder-feeder mechanism. In Section 3 we separate the results of the classification in seeding and non-seeding cases and compare them to a very simple visual detection. We present our conclusions

of this study in Sect. 4.





## 2 Methodology of the Arctic MLC classification algorithm

### 2.1 Datasets

Ny-Ålesund is located along a fjord on the west coast of the Arctic archipelago Svalbard (78.9 °N, 11.9 °E). Due to its location in the North Atlantic region of the Arctic, clouds above Ny-Ålesund are not only influenced by typical high-Arctic
stable weather conditions but are also frequently connected with cyclonic systems, as well as influenced by the mountainous orography of the archipelago. The occurring clouds might therefore differ from other Arctic sites, especially those over the pack ice. However, due to the good access to a one-year dataset of both radiosonde profiles and radar, it is a suitable choice for the evaluation of the detection algorithm.

For the classification radiosonde profiles and radar data from Ny-Ålesund between 10 June 2016 - 9 June 2017 are analysed.
Out of this 1-year period we analyse 278 days when both radiosonde and radar data is available. We consider the height range between 0 and 10 km. For each day, only the time frame of one hour after the radiosonde launch was considered. The regular launch time for the Ny-Ålesund radiosondes is 11 UTC. During campaign periods (e.g. 5 - 20 December 2016), additional launches at 5, 17, and 23 UTC are available. Within the analysed 1-year period, the station has changed the operational radiosonde type from Vaisala RS92 (until 11 April 2017) to Vaisala RS41 (from 12 April 2017), respectively. The humidity
sensor of the RS92 (RS92, 2013) has a manufacturer given uncertainty of 5% and a response time of < 0.5 s to < 20 s (for + 20 °C to - 40 °C, 6 ms$^{-1}$, 1000 hPa), while the RS41 (RS41, 2017) is described with an uncertainty of 4% and a response time of < 0.3 s to < 10 s (for + 20 °C to - 40 °C, 6 ms$^{-1}$, 1000 hPa), respectively. The radiosonde data with 1 s resolution were applied from Sommer et al. (2012) for the RS92 period, and from Maturilli (2017) for the RS41 period. All radiosondes were launched on balloons with an ascent rate of approximately 5 ms$^{-1}$. The horizontal drift of the sondes depends on the
atmospheric wind conditions.

A 94-GHz Doppler radar has been operated in Ny-Ålesund since 10 June 2016 by the University of Cologne as part of the (AC)³ project ("Arctic Amplification: Climate Relevant Atmospheric and Surface Processes and Feedback Mechanisms"; Wendisch et al. (2017)). The vertical resolution of the radar is 20 m and it reaches up to a maximum height of 12 km. The radar measures continuously with a temporal resolution of 30 s. A detailed description is found in Küchler et al. (2017). The
radar reflectivity factor was corrected for gaseous attenuation and the calibration was done in the way that a cloud at 273 K containing $1 \times 10^{-6}$ m$^{-3}$ droplets of $D = 100$ µm has a reflectivity factor of 0 dBZ at all frequencies.

For the cloud classification as step 1, radiosonde profiles are analysed regarding ice-supersaturation and ice-subsaturation. Secondly, as step 2, radar data is included in order to verify the MLC occurrence in these super- and subsaturated layers.

### 2.2 Classification step 1: Potential MLCs and sublimation calculation based on radiosonde profiles

The classification is divided into a step 1 and a step 2, as illustrated in Fig. 1. In step 1 we identify ice super- and ice subsaturated layers in the radiosonde profiles and calculate if ice crystal seeding is possible between these layers. We use the relative humidity with respect to liquid water from the radiosonde profile in combination with the temperature measurement and the formula of Hyland and Wexler (1983) to calculate the relative humidity with respect to ice. A sensitivity study where



measurement uncertainties are accounted for by also considering the relative humidity $\pm$ 5 % is shown in Appendix Fig. A1 and Fig. A2. Super- and subsaturated layers are identified using a threshold of 100 % relative humidity with respect to ice. The same threshold was also chosen by Treffeisen et al. (2007). When using a different threshold, e.g. 120 %, the results do not change significantly. If the temperature at certain levels is above 0 °C, then relative humidity with respect to water is chosen for

limiting the subsaturated layer. Numerous very thin super- and subsaturated layers (< 100 m) exist in the radiosonde profiles, but these layers are too thin to be considered a relevant contribution to the described processes. In order to sort out some of these irrelevant layers, but also to include thin cloud layers (Luo et al., 2008), the minimum thickness limits for the supersaturated and subsaturated layers are set to 100 m. This is in close agreement to Verlinde et al. (2007) finding layers to vary between 50 m to 300 m in depth. In order to detect a potential MLC, the criteria of detection is one *subsaturated layer in between*

(in the following termed cloudfree layer) one *supersaturated layer just above* (cloud layer) and one *supersaturated layer just below* (cloud layer). Subsaturated layers between two supersaturated layers at temperatures above 0 °C are not considered, as they are not relevant for our main point of focus, ice crystal seeding. Note that this means that we might underestimate the amount of multilayer clouds. If there is no supersaturated layer or only one single supersaturated layer, then these cases are not considered further for MLC detection (dark blue and green case in Fig. 1).

In the next step the sublimation calculation is done in order to answer if a falling ice crystal could survive its path through the subsaturated layer. For this the equation of vapour deposition is used to calculate the reduction of ice crystal mass due to sublimation (ice to vapour),

$$\frac{dm}{dt} = 4\pi C \rho_i G_i s_i \tag{1}$$

$$G_i = \left[ \frac{\rho_i R T}{M_w D_v e_i} + \frac{\rho_i l_s}{M_w k_T T} \left( \frac{l_s}{RT} - 1 \right) \right]^{-1}. \tag{2}$$

$m$ is the mass of one ice crystal and $C$ is its capacitance (Lamb and Verlinde, 2011). The capacitance replaces the radius $r$ of a liquid sphere and takes the shape of the ice crystal into account. A simplified approach is to use $C = \pi/2 \cdot r$ for a hexagonal plate (Seifert and Beheng, 2006; Harrington et al., 1995). $\rho_i$ is the density of ice, $G_i$ the growth parameter and $s_i$ the supersaturation

regarding ice, which is given by

$$s_i = \frac{e_i}{e_{sat,i}(T)} - 1 \tag{3}$$

and relates the actual ice saturation $e_i$ to ice equilibrium saturation at a given temperature $e_{sat,i}(T)$. In the case of subsaturation, the supersaturation is less then 0. Further variables in equation 2 are the temperature $T$, the heat transport $k_T$, the latent heat of sublimation $l_s$, the universal gas constant $R$, the molecular mass of water $M_w$ and the diffusion coefficient $D_v$. $D_v$ is in $\mathrm{m^2\,s^{-1}}$ and is calculated using

$$D_v = 0.211 \left( \frac{T}{T_0} \right)^{1.94} \frac{p_0}{p} \cdot 1 \times 10^{-4} \tag{4}$$




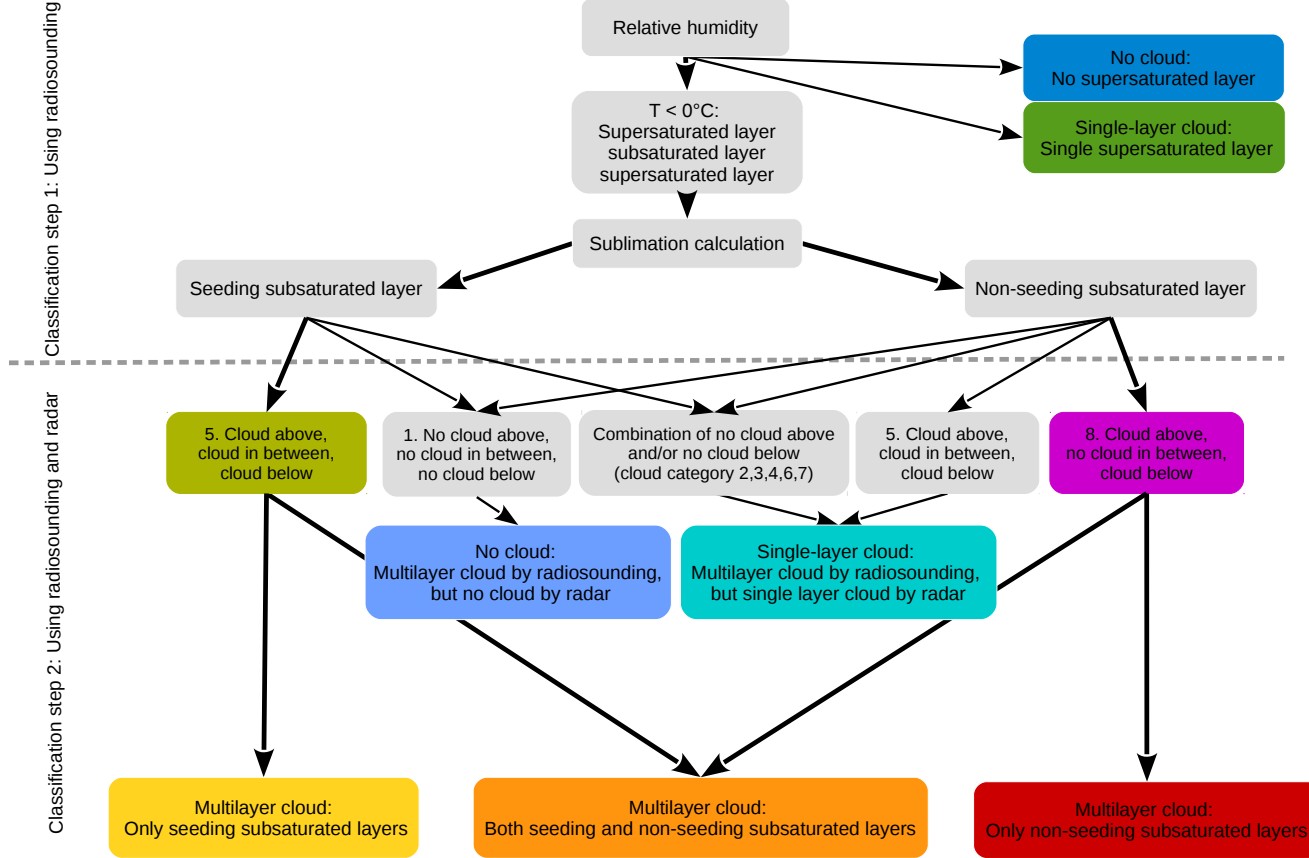

**Figure 1.** Overview of classification schemes: First only radiosonde data is used to detect one *subsaturated layer in between* two supersaturated layers, one just *above* and one just *below*. If this combination is found, then for the subsaturated layer the calculation of sublimation leads to seeding or non-seeding cases. Liquid layers above 0 °C are not considered. In the next step radar reflectivity factor data is added in order to detect cloud occurrence inside the investigated *supersaturated layer above* (*cloud above*), *subsaturated layer in between* (*cloud in between*) and *supersaturated layer below* (*cloud below*). The cloud category 5 'cloud above, cloud in between, cloud below' is counted as seeding MLC since it is most likely the seeding resulting in a radar signal in the subsaturated layer in between the cloud layers. The colours yellow, orange and red represent the resulting MLC categories.

with $T_0 = 273.15$ K and $p_0 = 1013.25$ Pa (Hall and Pruppacher, 1976). By using equation 1 the change of mass $dm$ with time $dt$ is obtained. In addition to that, assuming the radius-volume relation of a sphere by using $V = m/\rho_i$, a new radius is obtained at each time step by $r = \sqrt[3]{\frac{3V}{4\pi}}$. In order to answer the question if an ice crystal could reach the lower supersaturated layer, the mass, reduced due to sublimation, at each time step is combined with a fall speed in order to yield a fall distance. For this the fall speed $v(m)$ parametrisation of Seifert and Beheng (2006)

$$v(m) \cong \alpha \cdot m^\beta \left( \frac{\rho_{\text{air},0}}{\rho_{\text{air}}} \right)^\gamma \tag{5}$$



is used. Here are $\alpha = 0.217 \ \mathrm{m \, kg^{-\beta}}$, $\beta = 0.363$, $\gamma = 1/2$ empiric constants for cloud ice (hexagonal plates) and $\rho_{\mathrm{air,0}}$ is $1.225 \ \mathrm{kg \, m^{-3}}$. The air density $\rho_{\mathrm{air}}$ is given by $\rho_{\mathrm{air}} = p/(R_{\mathrm{s}} \cdot T)$, where $p$ is the actual pressure and $R_{\mathrm{s}}$ is the specific gas constant of air. The calculation is done using the forward Euler method and a time step of 0.01 s. The initial ice crystal size is assumed to be $r = 100$ µm, but also $r = 50$ µm and 150 µm are evaluated. Mean conditions of pressure, temperature and

humidity of each analysed subsaturated layer are used. If the ice crystal survives until the lower supersaturated layer, then it is called a **seeding** subsaturated layer. A **non-seeding** subsaturated layer means that the given ice crystal does not reach the lower next supersaturated layer because it sublimates completely.

As an example for the classification we show the classification for the case on 3 November 2016 in Fig. 2a. There are four subsaturated layers regarding ice and these are indicated by red horizontal lines. For the subsaturated layer 1 between

4.26 km and 3.85 km height the sublimation calculation is shown in Fig. 3. In Fig. 3a the change of mass and the calculated fall speed is shown and in Fig. 3b the resulting fall distance inside the subsaturated layer 1 is shown. An ice crystal of initial size $r = 100$ µm will sublimate completely before reaching the lower supersaturated layer and therefore the subsaturated layer 1 is a non-seeding one (red line in Fig. 3b). In the case of $r = 150$ µm the subsaturated layer 1 would be a seeding case. In the subsaturated layers 2, 3 and 4 an ice crystal of initial size $r = 100$ µm will survive and these layers are therefore determined as

seeding layers (sublimation calculation not shown).

## 2.3   Classification step 2: Cloud occurrence based on radiosonde profiles and radar

The aim of adding radar data to the classification is to cross-check the subsaturated layers in the radiosonde profiles with actual cloud occurrence. We use the radar reflectivity factor $Z$ from the Doppler radar in Ny-Ålesund. The continuous radar data has to be averaged. Here the start time is chosen 30 minutes before the radiosonde launch and the end time is 30 minutes after the

radiosonde reached 10 km height. For the 3 November 2016 the evaluated time period of the radar data is visualised by black lines in Fig. 2b. The heights of the super- and subsaturated layers, derived from the radiosonde humidity measurement, are indicated by red horizontal lines in Fig. 2b. In the *supersaturated layer above* only the lowest part is of interest for potential ice crystal seeding, since from here the ice crystal might fall. We consider only the lowermost 100 m of this supersaturated layer. A measured radar reflectivity factor above the detection limit means that cloud droplets or ice crystals reflect the radar

signal back. No radar reflectivity factor data means that the detection limit of -67 dBZ at 100 m to -38 dBZ at 10 km was not reached (Küchler et al., 2017). The averaged radar data is evaluated regarding if more than 50 % of the datapoints contain radar reflectivity factor data (coloured in Fig. 2b) meaning that cloud droplets or ice crystals were present. If so it is defined as cloud by the algorithm. If less than 50 % of the data points contain radar reflectivity factor data (white in Fig. 2b), it is defined as no cloud. For the *subsaturated layer in between* only the lowest part is evaluated to address the question if the ice

crystal survives so far. The lowermost 100 m are considered, but if the layer is thinner than 100 m only the available vertical thickness is considered. Again, if more than 50 % contain radar reflectivity factor data, it is considered as cloud. If less than 50 % contain radar reflectivity factor data, it is considered as no cloud. In the *supersaturated layer below* any radar signal at any height is of interest for potential ice crystal seeding. As soon as the ice crystal reaches this supersaturated layer it has survived. The ice crystal does not decrease in size, so that it can influence a cloud no matter at which height it is within the





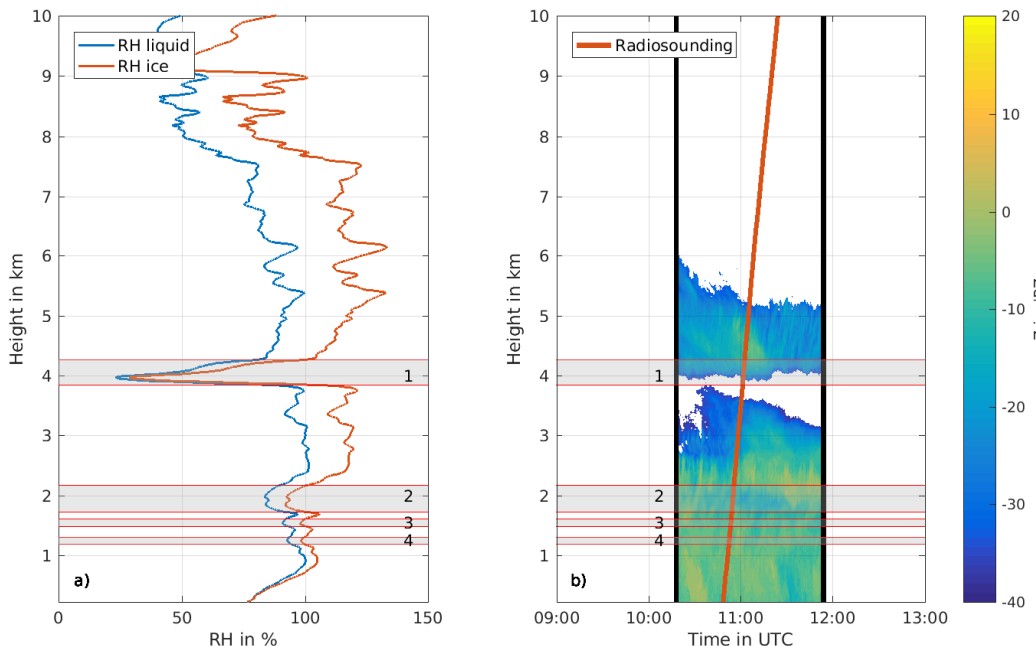

**Figure 2.** 3 November 2016 in Ny-Ålesund: **a**) Radiosonde profile between 10:48 -11:24 UTC (0 and 10 km height). Relative humidity (RH) with respect to water in blue and relative humidity with respect to ice in red. **b**) Radar reflectivity factor $Z$. The red vertical line visualises the ascend of the radiosonde. The black vertical lines visualise the time period considered for analysing the radar data. The red horizontal lines and the numbers 1,2,3 and 4 visualise the subsaturated layers. The grey colour visualises the subsaturated layers. At the time when the radiosonde reached the supersaturated layer 1 at 3.85 km due to horizontal wind drift the radiosonde is 3.68 km away from the radar.

supersaturated layer. For the *supersaturated layer below* the algorithm, starting from the top, searches for any layer of 100 m containing more than 50 % radar reflectivity factor data. If no layer of 100 m containing more than 50 % radar reflectivity factor data is found, at the lower boundary of this supersaturated layer the evaluated vertical thickness is decreased until 20 m. If no layer contains more than 50 % radar reflectivity factor data, it is considered that no cloud is present in this layer (*no cloud*).

5     In the example of 3 November 2016 (Fig. 2) the *supersaturated layer above* the subsaturated layer 1 is cloud containing (*cloud above*), the subsaturated layer 1 is not cloud containing (*no cloud in between*) and the *supersaturated layer below* is cloud containing (*cloud below*). The classification sorts the 3 November 2016 as MLC case. Analysing each combination of *supersaturated layer above*, *subsaturated layer in between* and *supersaturated layer below* results in the eight different cloud categories presented in Table 1.

10     For the non-seeding cases the classification considers the cloud category 8 (*cloud above*, *no cloud in between*, *cloud below* ▄▄) as MLC (purple and red in Fig. 1). We here refer to the MLC definition of two separate clouds with clear visible interstice in between (Liu et al., 2012). Since seeding ice crystals result in a signal in the radar data, it is difficult to distinguish this radar signal from the radar signal caused by cloud particles (Verlinde et al., 2007, 2013). The classification includes therefore for





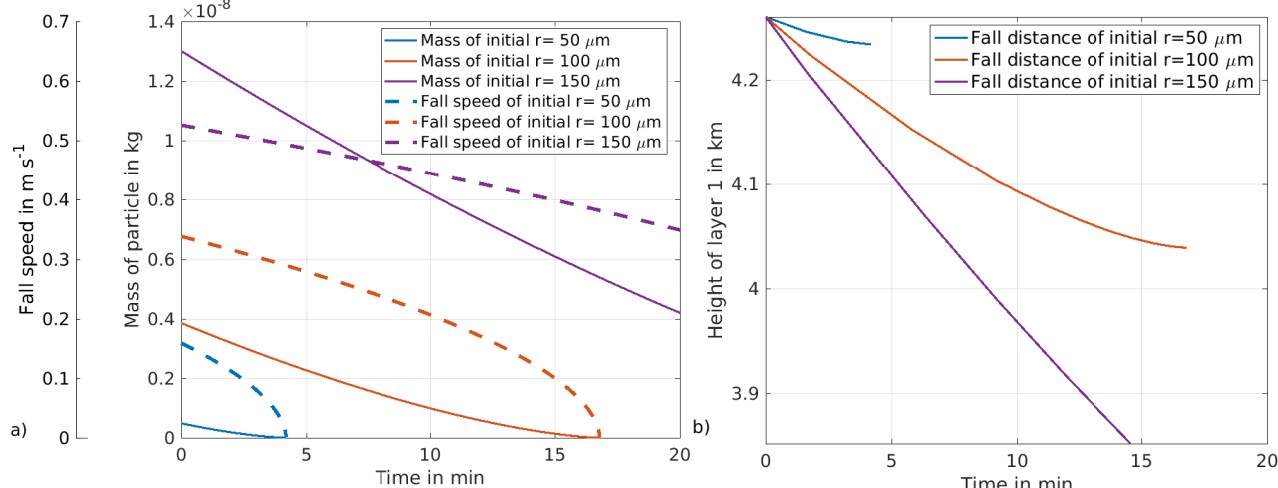

**Figure 3.** Calculation of sublimation for the layer 1 between 4.26 km and 3.85 km height at 3 November 2016: **a)** Fall speed and change of mass of ice crystal with time. **b)** Fall distance of ice crystal with time. The evaluated initial ice crystal sizes are $r = 50\,\mu\text{m}$, $100\,\mu\text{m}$ and $150\,\mu\text{m}$.

**Table 1.** Overview of the classification into eight different cloud categories. $N$ means no cloud and $C$ means cloud. SLC means single-layer cloud, MLC means multilayer cloud.

| | Cloud category # | | | | | | | |
|---|---|---|---|---|---|---|---|---|
| | 1 | 2 | 3 | 4 | 5 | 6 | 7 | 8 |
| above | $N$ | $N$ | $N$ | $N$ | $C$ | $C$ | $C$ | $C$ |
| in between | $N$ | $N$ | $C$ | $C$ | $C$ | $C$ | $N$ | $N$ |
| below | $N$ | $C$ | $C$ | $N$ | $C$ | $N$ | $N$ | $C$ |
| | **no cloud** | **SLC** | **SLC** | **SLC** | **seeding MLC** | **SLC** | **SLC** | **non-seeding MLC** |

the seeding cases the cloud category 5 (*cloud above*, *cloud in between*, *cloud below* ) as MLC (light green and yellow in Fig. 1). Therefore the classification's result should be treated as upper limit for MLC occurrence. A multilayer cloud containing several subsaturated layers of which some can be seeding and some non-seeding (at least one of each kind) is sorted as own multilayer category (orange in Fig. 1). The 3 November 2016 is an example to this category since layer 1 is a non-seeding layer and the layers 2, 3 and 4 are seeding layers. The classification sorts the cloud category 1 (*no cloud above*, *no cloud in between*, *no cloud below* ) as no cloud (light blue in Fig. 1). The cloud categories 2 ( ), 3 ( ), 4 ( ), 6 ( ), 7 ( ) are sorted as single-layer cloud (turquoise in Fig. 1). In the following section we show the results given by our classification for the one-year dataset used for the analysis.



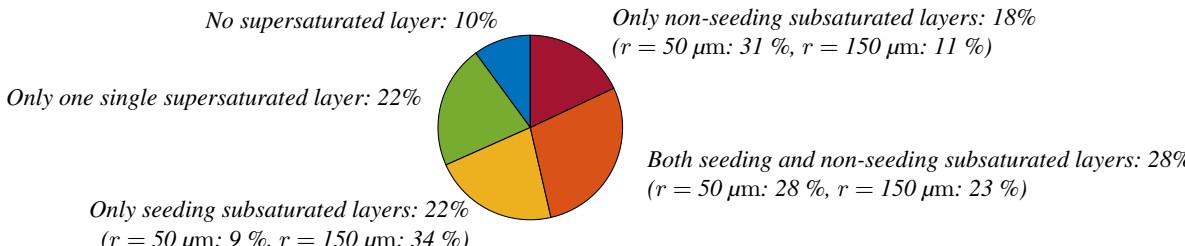

No supersaturated layer: 10%

Only non-seeding subsaturated layers: 18%
($r = 50\,\mu$m: 31 %, $r = 150\,\mu$m: 11 %)

Only one single supersaturated layer: 22%

Both seeding and non-seeding subsaturated layers: 28%
($r = 50\,\mu$m: 28 %, $r = 150\,\mu$m: 23 %)

Only seeding subsaturated layers: 22%
($r = 50\,\mu$m: 9 %, $r = 150\,\mu$m: 34 %)

**Figure 4.** Classification step 1 using an initial ice crystal size of $r = 100$ μm: Relative occurrence of supersaturated layers and seeding and non-seeding subsaturated layers. 100 % equals 278 relative humidity profiles. Percentages in brackets refer to the calculation using the initial ice crystal sizes $r = 50$ μm and 150 μm. For the categories 'no supersaturated layer' and 'only one single supersaturated layer' there are no changes in percentage. The values are rounded to zero decimal places.

## 3   Results and discussion of the classification applied to the Ny-Ålesund dataset

### 3.1   Results of classification step 1

The classification step 1 evaluates relative humidity profiles in order to detect seeding and non-seeding subsaturated layers. For the sublimation calculation primarily an initial ice crystal size of $r = 100$ μm is used. The result is presented in Fig. 4. The

criteria for potential MLC detection in classification step 1 is the combination of a *supersaturated layer above*, a *subsaturated layer in between* and a *supersaturated layer below*. This combination occurs in 68 % of the profiles (22 % yellow + 28 % orange + 18 % red in Fig. 4), which means that in 68% of the analysed radiosonde profiles we find potential MLCs. The possibility of microphysical interaction by seeding exists in 50 %. Varying the initial ice crystal size has a large, non-linear impact on the distribution between seeding and non-seeding subsaturated layers. A smaller initial ice crystal size leads to more non-seeding

layers, a larger initial ice crystal size leads to more seeding layers (numbers in brackets in Fig. 4). A seasonal cycle (Fig. 5) in this one-year dataset is not visible.

### 3.2   Results of classification step 2

In many cases of the 68 % potential MLC occurrence the results obtained by the radiosonde profiles disagree with actual MLC occurrence observed by the radar. In order to cross-check actual cloud occurrence in the *supersaturated layer above*,

in the *subsaturated layer in between* and in the *supersaturated layer below*, radar data is included in the classification step 2. Including radar data leads to eight different cloud categories (Sect. 2.3). The cloud categories are then separated regarding if the *subsaturated layer in between* is a non-seeding or seeding case according to step 1. For the non-seeding cases the cloud categories are shown in Fig. 6. The cloud category 8 (*cloud below*, *no cloud in between*, *cloud above*) is counted as MLC and therefore coloured purple. All other cloud categories occurring (1,2,3,5,7) are not considered as MLCs and are therefore

coloured dark grey and light grey. There is a high amount of the cloud category 1 (*no cloud below*, *no cloud in between*,



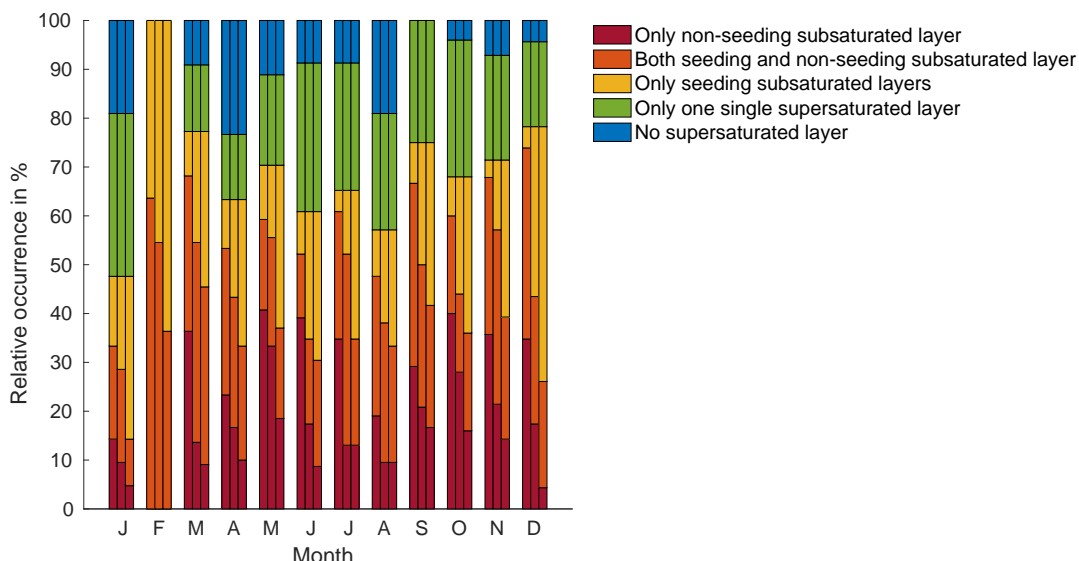

**Figure 5.** Temporal distribution of MLC days using classification step 1. For each month the left bar refers to the initial ice crystal size $r = 50$ μm, the middle bar refers to the initial ice crystal size $r = 100$ μm and the right bar refers to the initial ice crystal size $r = 150$ μm.

*no cloud above*, dark grey in Fig. 6). This means no cloud is visible in the radar even though a potential MLC was detected in the radiosonde profiles. Even if the layers *above* and *below* are supersaturated with respect to ice, the lack of suitable IN can prevent cloud formation (dark grey in Fig. 6 and Fig. 7). Therefore including radar data is of importance to the MLC classification. Ice-supersaturation without cloud formation is a global phenomena in the upper troposphere and does also occur

5 in the Arctic (Spichtinger et al., 2003). Spichtinger et al. (2002) explains this ice-supersaturation without cloud formation with the lack of aerosol as IN. Also for the cloud categories 2 (*no cloud above*, *no cloud in between*, *cloud below*, dark grey in Fig. 6) and 7 (*cloud above*, *no cloud in between*, *no cloud below*, dark grey in Fig. 6) too little IN can explain the missing cloud formation in the supersaturated layers. For cloud category 2 in the seeding case additionally the formation of seeding ice crystals is prevented (dark grey in Fig. 7). Besides the possible lack of IN also horizontal drift of the radiosonde away from the

10 radar and inaccuracies due to time averaging of the radar data can explain contradictions between radiosonde profiles and radar (light grey in Fig. 5 and Fig. 6). This leads to the cloud categories 3 (*no cloud above*, *cloud in between*, *cloud below*, light grey in Fig. 6) and 5 (*cloud above*, *cloud in between*, *cloud below*, light grey in Fig. 6) where a cloud signal is measured inside the subsaturated layer and these cloud categories are not counted as MLCs.

For the seeding subsaturated layers the cloud categories are shown in Fig. 7. The cloud category 8 (*cloud below*, *no cloud*

15 *in between*, *cloud above*) does not occur. This is explained by the fact that seeding ice crystals will make a signal in the radar reflectivity data. The cloud category 6 (*cloud above*, *cloud in between*, *no cloud below*) is a seeding case, but since the lower cloud layer is missing, no MLC case. The cloud categories 1,2,7, where missing cloud activation can be explained by missing





IN, are coloured dark grey. The cloud category 5 (*cloud above*, *cloud in between*, *cloud below*) is coloured light green in Fig. 7 and is considered as seeding MLC. For distinguishing a seeding MLC from a single-layer cloud a lidar/ceilometer detecting multiple cloud layers would be needed.

In Fig. 8 the result of the cloud classification step 2 using both radiosonde profiles and radar is presented. MLCs occur in 29 % of the investigated profiles (6 % 'only non-seeding', red + 3 % 'both seeding and non-seeding', orange + 20 % 'only seeding' MLC, yellow). Single-layer clouds occur in 50 % of the investigated profiles (28 % 'multilayer cloud by radiosounding, but single-layer cloud by radar', turquoise + 22 % 'single-layer clouds by radiosounding', green). No cloud layer occurs in 22 % of the investigated profiles (12 % 'multilayer cloud by radiosounding, but not cloud by radar', light blue + 10 % 'no cloud by radiosounding', dark blue). A seasonal variation (Fig. 9) in between months in this one-year dataset is very weak for the MLC categories ('only non-seeding multilayer clouds', 'both seeding and non-seeding multilayer clouds' and 'only seeding multilayer clouds'). There is a slight increase in MLC occurrence between July and November and February to March.

The impact of different ice crystal sizes used in classification step 2 is presented as numbers in brackets in Fig. 8 and as bars in Fig. 9 . The main impact is that for a smaller ice crystal there are less 'only seeding multilayer cloud' cases and more 'multilayer cloud by radiosounding, but single-layer cloud by radar' cases. This is explained by the cloud category 5 ('*cloud above*, *cloud in between*, *cloud below*') occurring frequently and sorted as MLC in the seeding cases and as single-layer cloud in the non-seeding cases. Because of this different sorting of seeding and non-seeding cases, the impact of the ice crystal size is less strong in classification step 2 compared to step 1.

A sensitivity how the results would change assuming an uncertainty of the radiosonde humidity of $\pm$ 5 % is shown in Appendix Fig. A1 and Fig. A2. The measurement uncertainties lead to variations in the results of the same order of magnitude as when varying the ice crystal size. If the relative humidity is on average overestimated, the impact on the results is of smaller importance than if the relative humidity is on average underestimated. This might be explained by the minimum thickness threshold of 100 m used for identifying supersaturated and subsaturated layers limiting the effect when overestimating the relative humidity.

### 3.3 Discussion and evaluation of the results using skill scores

For evaluating the MLC occurrence derived by the classification steps 1 and 2 skill scores are used. First classification step 1 (using only radiosonde data) is compared to classification step 2 (using both radiosonde and radar). Secondly classification step 2 is compared to a visual inspection. The visual inspection is done manually. We inspect the radar images and decide whether it is a visual MLC or no visual MLC. For the visual inspection we consider a shorter time periode like that of the radiosonde ascent rather than the average over one hour like the detection algorithm does. Small cloud stains are not counted as clouds and clouds containing small cloud free holes are counted as clouds.

The variables $A$,$B$,$C$,$D$ needed for deriving the skill scores are given as in Table 2. Out of these variables the probability of detection $POD$ is defined as

$$POD = \frac{A}{A+C} \tag{6}$$





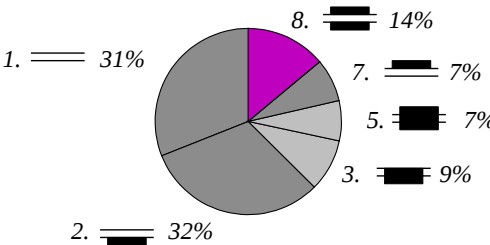

**Figure 6.** Non-seeding cases: Cloud categories of all non-seeding subsaturated layers. *In between* refers to the subsaturated layer. *Above* and *below* refers to the supersaturated layers above or below the subsaturated layer. 100 % equals all non-seeding subsaturated layers. Non-seeding is calculated using an ice crystal of the size $r = 100\,\mu\text{m}$. The values are rounded to zero decimal places.

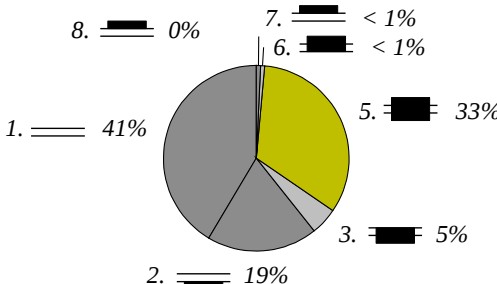

**Figure 7.** Seeding cases: Cloud categories of all seeding subsaturated layers. *In between* refers to the subsaturated layer. *Above* and *below* refers to the supersaturated layers above or below the subsaturated layer. 100 % equals all seeding subsaturated layers. Seeding is calculated using an ice crystal of the size $r = 100\,\mu\text{m}$. The values are rounded to zero decimal places.

and shows perfect detection at $POD = 1$ and no detection at $POD = 0$.

The false alarm rate $FAR$ is defined as

$$FAR = \frac{B}{A+B} \tag{7}$$

and gives $FAR = 0$ for no false alarms and $FAR = 1$ for only false alarms.

The Heidke skill score $HSS$

$$HSS = 2\frac{AD - BC}{(A+C)(C+D) + (A+B)(B+D)} \tag{8}$$

5   evaluates the total predictability with values reaching from $HSS = -\infty$ to 1. $HSS = 0$ means that there is no predictability.

For the evaluation of classification step 1 (using only radiosonde) the variables $A,B,C,D$ are presented in Table 3. There the results of classification step 1 are divided into MLC and no MLC. MLCs in classification step 1 are defined as one *supersaturated layer above*, one *subsaturated layer in between* and one *supersaturated layer below*. If a MLC is detected by



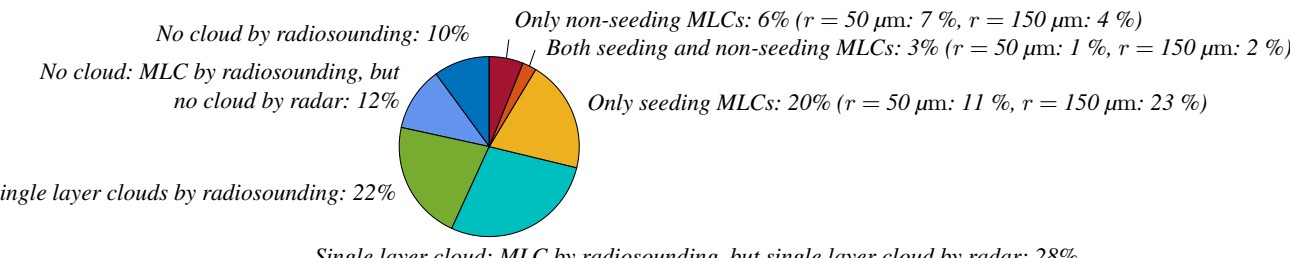

**Figure 8.** Cloud occurrence derived from using both radiosonde and radar for detection. For the categories the same colours as in Fig. 1 are used. 100 % equals 278 days (analysed days within the one-year data set). Seeding and non-seeding is calculated using an ice crystal of the size $r = 100\,\mu$m. Percentages in brackets refer to the calculations using different initial ice crystal sizes $r = 50\,\mu$m and $150\,\mu$m. The values are rounded to zero decimal places.

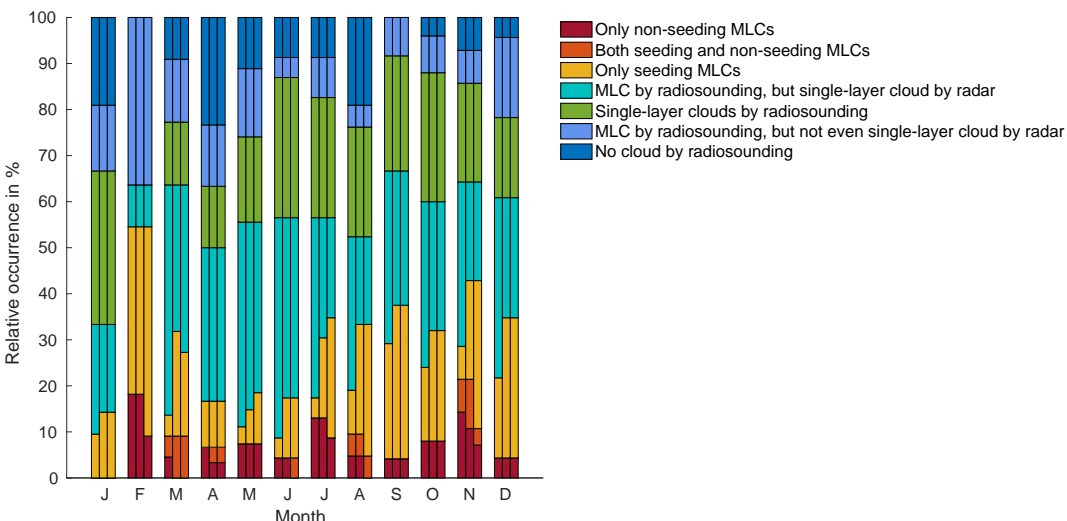

**Figure 9.** Temporal distribution of MLC days using classification step 2. For each month the left bar refers to the initial ice crystal size $r = 50\,\mu$m, the middle bar refers to the initial ice crystal size $r = 100\,\mu$m and the right bar refers to the initial ice crystal size $r = 150\,\mu$m.

classification step 1, the best estimate for evaluation is given by classification step 2 (MLC or no MLC by radar). If no MLC is detected by classification step 1, the best estimate for evaluation is done by the manual visual inspection of the radar images. The manual visual inspection is necessary owing to the non-existence of classification step 2 if there is no MLC by radiosounding. In Table 4 the resulting skill scores are shown. The good $POD$ of 0.99 (for $r = 100\,\mu$m) affirms that there is

5   no big loss of MLC cases when applying classification step 1 (only 0.4 % for $r = 100\,\mu$m). Varying the initial ice crystal size to $r = 50\,\mu$m and $r = 150\,\mu$m does almost not cause any variation of $POD$. This means the impact of chosen initial ice crystal





**Table 2.** Skill score evaluation: Definitions of the evaluation variables $A,B,C,D$ used for the evaluation of the MLC occurrence derived by the classification steps 1 and 2 in comparison to a best estimate of MLC occurrence.

|  |  | Best estimate | |
|---|---|---|---|
|  |  | MLC | no MLC |
| Classification | MLC | $A$ | $B$ |
|  | no MLC | $C$ | $D$ |

size on the predictability is limited. $FAR$ being 0.58 reveals that about half of the MLC estimated from radiosonde humidity measurements is 'no MLC by radar'. Therefore classification step 1 represents a reliable upper limit (68.4 %) for identifying MLC days, but the actual number might be as low as less than the half (28.8 %). This limited predictability leads to a low Heidke skill score of $HSS = 0.31$ for $r = 100\,\mu m$.

**Table 3.** Evaluation of the MLC results of radiosonde detection (classification step 1) in comparison to radar detection (classification step 2). 'MLC by radar' is given by cloud category 8 for the non-seeding cases and by cloud category 5 for the seeding cases. The evaluation is done for the ice crystal sizes $r = 50\,\mu m$, $100\,\mu m$ and $150\,\mu m$. The values are rounded to one decimal place.

|  | $r = 50\,\mu m$ | | $r = 100\,\mu m$ | | $r = 150\,\mu m$ | |
|---|---|---|---|---|---|---|
|  | MLC by radar | no MLC by radar | MLC by radar | no MLC by radar | MLC by radar | no MLC by radar |
| MLC by radiosounding | 19.8 % | 48.6 % | 28.8 % | 39.6 % | 29.1 % | 39.2 % |
| no MLC by radiosounding | 0.4 % | 31.3 % | 0.4 % | 31.3 % | 0.4 % | 31.3 % |

**Table 4.** Skill scores for comparison of MLC results of radiosounding and radar. The skill scores are calculated for the ice crystal sizes $r = 50\,\mu m$, $100\,\mu m$ and $150\,\mu m$. The values are rounded to two decimal places.

|  | $r = 50\,\mu m$ | $r = 100\,\mu m$ | $r = 150\,\mu m$ |
|---|---|---|---|
| $POD$ | 0.98 | 0.99 | 0.99 |
| $FAR$ | 0.71 | 0.58 | 0.57 |
| $HSS$ | 0.20 | 0.31 | 0.31 |

5    Next we evaluate classification step 2 and the results are presented in Table 5 and Table 6. Due to the missing possibility to distinguish falling ice crystals from cloud particles in the radar image, including seeding to our classification leads to high uncertainties. Therefore for evaluating classification step 2, we only consider the non-seeding MLCs (cloud cat. 8). This is a similar approach as done by Intrieri et al. (2002), who defined MLCs as two separate clouds with clear visible interstice in between. For the evaluation of classification step 2 we use the manual visual inspection as best estimate. Also for the manual

10   visual inspection we do not account for the possibility of seeding, meaning that we count a connected radar signal in the vertical as single-layer cloud.





Classification step 2 classifies 8.3 % MLCs ($r = 100\,\mu$m in Tab. 5). This represents a lower limit for identifying MLC days, since classification step 2 is not able to classify 10.1 % (4.0 % are classified as seeding MLC and 6.1 % as no MLC). The actual number of MLCs might therefore be twice as high (8.3 % + 10.1 % = 18.4% for $r = 100\,\mu$m). This limited probability is underlined by $POD$ being 0.45 (Tab. 6). Problems of the classification are given by the not exact accordance between radiosonde profile and radar. While the radiosonde ascents, it is horizontally drifted away from the radar by wind. Additionally the radar measurements have to be averaged over time and this is not done by the visual inspection. An existing cloud, which is too weak or too short lasting in the radar image, can therefore lead to discrepancies between the classification and the visual detection. A too high cloud top or base compared to the relative humidity threshold, a missing relative humidity layer or too many relative humidity layers in a not changing radar image do also cause erroneous classification.

However, few false alarms (0.4 % for $r = 100\,\mu$m) cause a low $FAR$ of 0.04. This reveals predictability by a $HSS$ skill score of 0.56. A larger radius ($r = 150\,\mu$m) leads to more seeding cases. This results in less non-seeding MLCs accounted for in classification step 2 and worsens therefore the predictability for $r = 150\,\mu$m ($HSS = 0.47$) in comparison to $r = 100\,\mu$m ($HSS = 0.56$). Using the smaller radius of $r = 50\,\mu$m does not change the results. Even if there is less seeding, these cases belong to the category 'both seeding and non-seeding' and do therefore not change the results. In classification step 1 the larger radii $r = 100\,\mu$m and $r = 150\,\mu$m lead to the best Heidke skill score ($HSS = 0.31$) in comparison to the small radius. However, in classification step 2 the smaller radii $r = 50\,\mu$m and $r = 100\,\mu$m lead to the best Heidke skill score ($HSS = 0.56$) in comparison to the large radius $r = 150\,\mu$m. In this way we decided to focus on the radius $r = 100\,\mu$m in the previous sections of this manuscript.

**Table 5.** Evaluation of the MLC results including only the non-seeding MLC of the classification step 2 in comparison to manual visual detection. 'Non-seeding MLC' includes 'only non-seeding' and 'both seeding and non-seeding' MLC. 'Seeding MLC and no MLC' includes seeding MLCs, single-layer clouds and no cloud layers. The evaluation is done for the ice crystal sizes $r = 50\,\mu$m, $100\,\mu$m and $150\,\mu$m. The values are rounded to one decimal place.

|  | $r = 50\,\mu$m | | $r = 100\,\mu$m | | $r = 150\,\mu$m | |
|---|---|---|---|---|---|---|
|  | visual MLC | no visual MLC | visual MLC | no visual MLC | visual MLC | no visual MLC |
| non-seeding MLC | 8.3 % | 0.4 % | 8.3 % | 0.4 % | 6.5 % | 0.0 % |
| seeding MLC and no MLC | 10.1 % | 81.3 % | 10.1 % | 81.3 % | 11.9 % | 81.7 % |

**Table 6.** Skill scores for comparison of MLC results of the non-seeding cases of the classification step 2 and the visual detection. The skill scores are calculated for the ice crystal sizes $r = 50\,\mu$m, $100\,\mu$m and $150\,\mu$m. The values are rounded to two decimal places.

|  | $r = 50\,\mu$m | $r = 100\,\mu$m | $r = 150\,\mu$m |
|---|---|---|---|
| $POD$ | 0.45 | 0.45 | 0.35 |
| $FAR$ | 0.04 | 0.04 | 0.00 |
| $HSS$ | 0.56 | 0.56 | 0.47 |





## 4 Conclusions

In this work we use in-situ profiling by radiosondes and ground-based remote sensing by radar to identify Arctic MLCs between 0 - 10 km height. We evaluate relative humidity profiles regarding an ice-subsaturated layer in between two ice-supersaturated layers. This combination occurs in 68.4 % out of 278 analysed days (only one hour each day is analysed) using the minimum

considered thickness for the supersaturated and subsaturated layers of 100 m. A high amount of supersaturated layers found in the radiosonde profiles does not coincide with observed cloud occurrence, probably due to missing cloud activation. Only using radiosonde profiles is not sufficient for the detection of clouds. Therefore the classification is expanded by using radar data for excluding non relevant cases. The extended classification leads to 29 % MLCs with a very weak seasonal cycle. We investigate these MLC further regarding the possibility of seeding, which means if an ice crystal of the size $r = 100$ µm can

survive sublimation in the subsaturated layer when falling through this layer. We find that seeding can potentially occur in 23 % of the 278 investigated days. In these cases there is a radar signal in the subsaturated layer in between the two cloud layers. Here it remains as an unsolved question if this is actually due to seeding (falling ice crystals in between the two cloud layers) or due to one continuous cloud layer. Since the percentage for potentially seeding is as high as 23 %, the importance of seeding on the lower cloud is not negligible. The effects of the seeding on the lower cloud could be an increase in cloud ice,

and thereby precipitation formation and cloud dissipation. In order to gain more information about the existence of these ice crystals, further measurements of e.g. lidar would be needed.

Non-seeding means that the subsaturated layer is too thick or too dry for the ice crystal to survive the sublimation. Non-seeding MLCs are visible in the radar as two separated cloud layers and this occurs in 9 % of the analysed days. While it could have been envisioned that falling ice crystals occur in low concentrations below the radar detection limit, we find that MLCs

visible separated in the radar are unable to interact through seeding following from our sublimation calculation. However, radiative interactions, like a weakening of the lower cloud in the existence of a higher cloud, can occur. These interactions are most likely not captured correctly by weather models. However, the 9 % occurrence implies that clearly separated MLCs should probably not be neglected in weather models.

Cloud detection by satellites is challenging in the Arctic, but Liu et al. (2012) found Arctic MLCs to occur between 17 -

25 % of the investigated time. However, since the minimum considered cloud thickness was as big as 960 m, they assumed their MLC amount most likely to be underestimated. In order to evaluate our classification we compare our results to a manual visual inspection of the radar observations. Since the seeding cases can not be separated from single-layer cloud cases and therefore cause uncertainties, the seeding cases are excluded in the evaluation. The evaluation results in non-seeding MLC occurrence of 9 % being a reliable lower limit. However, the Heidke skill score $HSS$ for prediction is only 0.56. Changing the

ice crystal size has only little impact on the results. Neither a smaller initial ice crystal size of $r = 50$ µm nor a larger initial ice crystal size of $r = 150$ µm leads to major improvements. Erroneous detection is often caused by super- and subsaturated layers identified in the radiosonde data not overlapping with the radar cloud top and base. Also non-relevant, often thin super- and subsaturated layers cause problems. Here the uncertainties in the relative humidity measurements and the chosen minimum



height limits have to be kept in mind when examining these disagreements. The manual visible inspection results in 18.4 % non-seeding MLC occurrence.

Using our ground-based classification leads to a MLC occurrence between 8 - 29 % for Ny-Ålesund. If and how much this number will differ at a more typical high Arctic location, with less cyclonic and orographic influence but rather stable conditions caused by sea ice, remains an unsolved question. We show that seeding is more frequently possible than non-seeding and always causes a signal in the radar. Therefore uncertainties remain when distinguishing MLC from single-layer clouds in radar images. While extensive modelling studies (e.g. Klein et al. (2009) and Ovchinnikov et al. (2014)) have dealt with single-layer Arctic clouds, we suggest that the more complex microphysics and radiative properties of MLCs and their changes due to aerosol and climate pertubations should be a focus of future research.

*Code availability.* The code for the seeding/non-seeding multilayer cloud detection algorithm was written in Matlab and is available as supplement.

*Data availability.* The radiosonde data is available through Sommer et al. (2012) and Maturilli (2017). The radar is part of the (AC)[3] project and the data was provided by Kerstin Ebell.





# Appendix A

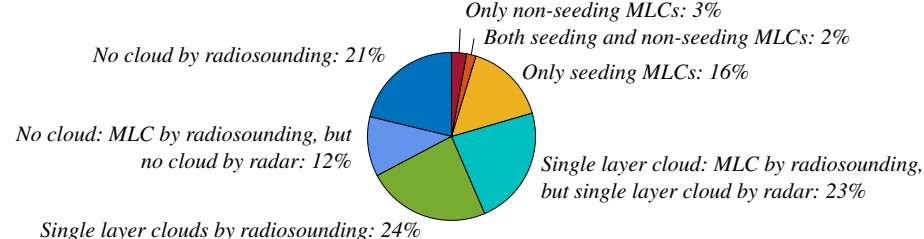

**Figure A1.** Cloud occurrence derived from using both radiosonde and radar for detection. For the radiosonde data the measurement uncertainty is considered to be -5 % over the whole radiosonde profile. Seeding and non-seeding is calculated using an ice crystal of the size $r = 100$ μm. The values are rounded to zero decimal places.

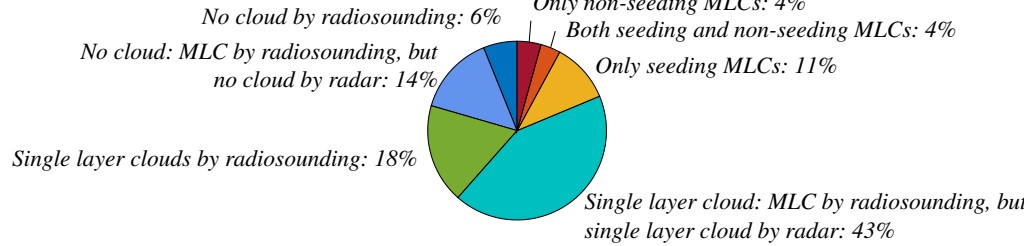

**Figure A2.** Cloud occurrence derived from using both radiosonde and radar for detection. For the radiosonde data the measurement uncertainty is considered to be +5 % over the whole radiosonde profile. Seeding and non-seeding is calculated using an ice crystal of the size $r = 100$ μm. The values are rounded to zero decimal places.



*Competing interests.* The authors confirm that they have no conflict of interest.

*Acknowledgements.* We gratefully acknowledge support by the SFB/TR 172 "ArctiC Amplification: Climate Relevant Atmospheric and SurfaCe Processes, and Feedback Mechanisms (AC)³ funded by the DFG (Deutsche Forschungsgesellschaft). In particular, the cloud radar observations are performed within the sub-project E02 of SFB/TR 172 and have been provided by Kerstin Ebell. The radiosonde data was
5  provided by the Alfred Wegener Institute. Luisa Ickes acknowledges the Swiss National Science Foundation (Early Postdoc.Mobility) for support.



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
