# Peer review of "Classification of Arctic multilayer clouds using radiosonde and radar data in Svalbard"

_Atmospheric Chemistry and Physics, 2018_

## Referee Comment (RC1) · Anonymous Referee #1 · 28 Sep 2018

Review of "Classification of Arctic multilayer clouds using radiosoundings and radar data" by Vessel et al.

Recommendation: Might be acceptable for publication after mandatory revision

This paper analyzes a year of data collected by a radar and radiosoundings at Ny-Alesund and attempts to determine the frequency of occurrence of multi-layer clouds, and in the case of multi-layer clouds whether the cloud layer underneath is seeded by the cloud above. The subject matter is timely because the Arctic is currently warming quicker than other parts of the planet, yet models have a difficult time accurately predicting the amount of warming. Better knowledge of the properties of arctic clouds, and on what controls them, is necessary in order to improve these predictions: any paper that hence contributes to our data base on the phases, heights and geometrical char-

acteristics of clouds is beneficial. The paper is well written and the presented analysis easy to understand.

Nevertheless, I fear that the paper as currently written is quite misleading. There are so many uncertainties and problems with the analysis (which, in their defense, the authors do a good job of identifying) that I fear the results that come out of the paper are not terribly useful. However, I think if the data were presented in an alternate way the study could be of potential use and hence I am recommending major revision rather than rejection.

Step 1 of their analysis uses the radiosonde data to identify the presence of multi-layer clouds. However, even though the probability of detection of the multi-layer clouds is 99% with this method, the false alarm rate of 58% "reveals that about half of the MLC detected by radiosounding is no MLC by radar." Thus, it seems that the paper should be reworded to emphasize that the use of the radiosonde data on its own does not reliably identify the occurrence of MLCs, but can be used in combination with radar data to give information on the presence of MLCs. The authors acknowledge the unreliability of the radiosonde data on their own to identify MLCs as they state "even if the layers above and below are supersaturated with respect to ice, the lack of suitable IN can prevent ice cloud formation." They also stated that "the results obtained by the radiosonde profiles disagree with actual MLC occurrence observed by the radar."

The second major problem with the analysis presented is the reliance on the chosen ice crystal size to calculate which of the upper layers of MLCs is seeding a lower layer. As stated by the authors "varying the initial ice crystal size has a large, non-linear impact on the distribution between seeding and non-seeding subsaturated layers." Further, their calculations substantially underestimate the variance that the size of the seeding ice crystal size might have. In several studies of in-situ measurements of mixed-phase clouds, the ice crystal sizes have been much larger than the 150 micrometer size assumed in the calculations here. Further, the calculations assume a hexagaonal plate which is not representative of the shapes of ice crystals in mixed-phase clouds. For

Interactive
comment

example, Korolev et al. (1998) found that over 98% of ice crystals in mixed-phase arctic clouds had irregular shapes. Thus, the uncertainties will be much larger than those stated, and the stated uncertainties are already huge. And, the base size of 100 micrometers is probably much smaller than the size of particles that will be emanating from the upper layer.

Another potential problem could be the lack of colocation between the radiosondes (which can drift large distances in the background wind) and the radar, which is again noted by the authors: "horizontal drift of the radiosonde away from the radar and in-accuracies due to time averaging of the radar data can explain contradictions between radiosounding and radar." Was any effort made to consider the advection of air parcels measured by the radiosondes so that the radar data at an appropriate time could be used in the analysis (provided that the air parcel was within the radar view volume at some time)?

MORE DETAILED COMMENTS:

Abstract: At first reading, I was confused that the 9% and 23% of the cases mentioned because it did not add to 100%. Perhaps mention that other cases (of the 8 categories) are included to avoid confusion.

Page 1, Line 24: I find trying to differentiate between the terms multilayered clouds and multilayer clouds (MLC) very confusing! To me, that is the exact same word.

Page 3, line 6: Do you expect any diurnal cycle in the cloud properties that would mean that the derived statistics are not representative of the Arctic as a whole?

Page 3, line 10: I think data is plural, so it should be "data are" rather than "data is"

Page 3, line 28: What statistical test was applied to show that the results did not change significantly?

Page 7, line 9: Why is only the lowermost 100 m considered?

Page 10, line 11: Vali (200x?) has recommended that ice nucleating particles (INPs) rather than ice nuclei (IN) be used in order to standardize terminology. Recommend that you use INP rather than IN.

Page 15, line 1: The cloud layers can slope up and down frequently (in relatively short distances or times) and that can have a big impact on averaging. Was this taken into account?

Page 17, line 3: Perhaps I am not looking in the right place, but I cannot find the supplement being referred to in this statement.

---

## Referee Comment (RC2) · Anonymous Referee #2 · 23 Oct 2018

Review of acp-2018-774 ——————

Summary of the manuscript

The study titled "Classification of Arctic multilayer clouds using radiosonde and radar data" by Maiken Vassel et al. describes an algorithm for the classification of multi-layer cloud occurrence for a one year dataset in Ny Alesund, Svalbard based on radiosonde and vertically-pointing cloud radar observations. The classification is two-fold: Firstly, only the conditions for cloud occurrence based on radiosonde humidity profiles consisting of two supersaturated layers separated by a subsaturated layer are analyzed. The fall distance of a hexagonal ice crystal of 100 micron size before complete evaporation in the subsaturated layer are estimated. The subsaturated layers are then classified into two categories. The first category is called "seeding", referring to layers with a

vertical extent lower than the fall distance before complete ice crystal sublimation. The second category is called "non-seeding", referring to layers with a vertical extent higher than the fall distance before complete before complete ice crystal sublimation. These maximum possible occurrence frequencies for multi-layer cloud occurrence based on supersaturated layers as identified by radiosonde ascents are then verified by cloud radar reflectivity profiles obtained within 30min before radiosonde launch and 30min after the radiosonde has reached 10 km altitude. Multilayer mixed-phase cloud occurrence was found in 29% of the cases based on the combined radiosonde-cloud radar estimation. One of the main finding of the paper is that about 50% of the multilayer clouds estimated solely from radiosonde humidity profiles are not classified as such by the radar. - But the conclusion that radiosounding data is not sufficient for multi-layer cloud occurrence classification since not only humidity but also concentrations of ice nucleating particle (INP) and cloud condensation nuclei (CCN) are crucial is not made.

I would suggest the manuscript to be published after major revisions. The authors should address the following points:

Major comments ——————-

The literature review in the introduction should be extended. For example: p.1 line 20: Some more recent publications on Arctic mixed-phase cloud properties should also be included, for example Shupe 2011 (DOI: 10.1175/2010JAMC2468.1). In this paper. e.g., the occurrence of mixed-phase clouds at Arctic sites was found up to altitudes of 7-8 km.

p.2 line 18: Please give a broader and more detailed introduction on seeder-feeder mechanism and why it is important to consider it in the Arctic. Also, be more specific in which way the ice crystals falling from the upper cloud "influence" the lower cloud. Since this is a central question of this study, it has to be properly introduced.

The points made in various parts of the manuscript (especially Section 2.3) need to be more precise instead of using colloquial expressions like "surviving" ice crystals etc.

[Figure]

(e.g., also: p.6 line 24-29; p. 10 line 1-16, p.11, line 29-30, p.16 lines 18-20, etc.: very colloquial language).

p.4 line 1-14: Please include a conceptual sketch of which kind of cloud layers you are considering indicating minimum depth of the layers, minimum vertical spacing between two supersaturated layers with a subsaturated layer between, temperature restrictions. . .otherwise it is really hard to follow.

p.4: It is mentioned that a simplified approach is used to determine the capacitance for a hexagonal plate. (By how much) does the capacitance differ for different ice crystal shape assumptions (columns/dendrites/quasi-spherical spheres)? Also, why do you use a radius-volume relation of a sphere (p.5) if you consider hexagonal plates? As e.g. shown by Mitchell, JAS 1996, ice particle fall speed is a strong function of ice particle shape and density. Only assuming one particular ice crystal shape (hexagonal plates) is not sufficient for the fall distance estimation. Sublimation calculations should at least be repeated for two other ice particle shapes with very different fall speed characteristics such as columns and dendrites which would lead to very different fall distances. Additionally, please mention the influence of up- and downdrafts on ice particle fall velocity and thus fall distance.

p.6 line 24-29: Very colloquial language describing the radar reflectivity above the detection limit (p.6) - please rephrase. Please describe you averaging more in detail: I assume you refer to temporal averaging at each altitude? Please include a third Panel in Fig.2 showing the radar reflectivity sensitivity profile and the averaged reflectivity of the example case (for 50% data points within the considered time span).

p.8: I haven't seen a definition for a cloud case – how long of a gap in time is needed to refer to a scene having two separate clouds at one altitude– one radar profile (30s?) or a few minutes? Or is this not considered at all and averaging over time around RS launch is done in a way that separate cloud occurrences at one height are averaged into "one"?

p.10 lines 1-4: I do not fully agree with your conclusion that no cloud return in the radar data always means cloud-free conditions. There could be situations in which the sensitivity of the radar is not high enough (LWC and IWC too low). – I thus strongly suggest to also use available profiling lidar data (ceilometer) instead of only radar data to check for cloud occurrence in supersaturated conditions. Although the ceilometer suffers from full attenuation at sufficiently optically thick clouds, it will likely increase the number of detected cloud occurrences from ground-based remote sensing observations.

Moreover, in the Arctic frequently clouds occur at very low altitudes which might sometimes be below the lowest radar/lidar range gate. On page 16 (line 15-16) you mention that a lidar would be useful. I strongly suggest making use of the existing ceilometer data (https://www.awipev.eu/awipev-observatories/cloud-cover/) in your study.

Minor comments —————————-

Title: I suggest adding "in Svalbard" to narrow down the geographical range of the study. Also, since you are only considering T< 0°C, you can add "cold" clouds in the title.

p.4: For all variables in the equations the units should be included.

p.1 Line 10-13: It is unclear which kind of "deviations" you refer to – please specify more in detail.

p.1 line 16: it should be "improve" instead of "improved"

p.1 line 16: hydrometeor shape and density (and thus terminal fall velocity) are of great importance, too.

p.1 line 21: You could add that the typical structure of stratiform mixed-phase cloud with supercooled liquid top layer and precipitating ice points to heteorogeneous ice formation processes (add citation).

p.1 line 21: rather from the "remote sensing point of view"

p.1 line 22ff: "variable lidar signals inside a more or less continuous radar signal" sounds very imprecise – please rephrase and refer to "cloud profiles obtained with vertically-pointing instruments" or sth. similar

p.1 line 24: make it easier for the reader and put multilayered vs. multilayer in Italics or bold font

p.1 line 24: In which measurement is the interstice of multilayer clouds visible – in profiles of radar or lidar or both?

p.2 line 2: I suppose, you mean "at least" two clouds in different heights since there can be very low boundary layer clouds, midlevel clouds, and high-level clouds occurring simultaneously

p.2 line 20: Please modify the sentence since you look at one specific Arctic site (. . .occur at Ny Alesund not "the Arctic")

p.2 line 20: Why "thereby"?

p.2 line 22: it should be "ground-based remote sensing" measurements

p.2 line 23: What do you mean by "easily accessible"?

p.2 line 25: be more precise in your wording: Instead of "radar" it should be profiling/zenith-pointing Doppler cloud radar

p.3 line 20: Please give a rough estimate of horizontal drift of the sondes based on their GPS tracking.

p.3 line 23: Please also indicate the lowest radar range gate and mention that the cloud Doppler radar is zenith-pointing. You mention a vertical radar range gate resolution of 20m, is it really the same at all altitudes (i.e., all chirps) and was the RPG radar really operated in the same mode (with the same vertical and temporal resolution) over the entire year? 30s temporal resolution seems very low – please verify this temporal resolution. . .or are you using data already averaged to Cloudnet temporal resolution?

p.3 line 25: Please indicate typical values of attenuation correction for the 94GHz radar at Ny Alesund.

p.3 line 25: What do you mean by "at all frequencies"?

p.6 line 4: ice crystal size r refers to maximum dimension? Motivate the choices of ice crystal size of 50/100/150 microns by citing typical values found in Arctic clouds.

p.6 line 4: It is unclear what you mean by "mean conditions": Mean over one hour after radiosonde launch?

p.6 line 5: "survive" is very colloquial, please replace or describe what you mean. Please change accordingly throughout the text.

p.6 line 17: The way it is presented it sounds like as if the radar is used to test for cloud occurrence (above/between/below) in general and not only for cloud occurrence in general (Also see Table 1). (?)

p.6 line 19-20: Clarify why you use 30min after the radiosonde reached 10km as end time and not e.g. simply a one hour time around RS launch (start 30min before and end 30min after launch)?

p.6 line 29-30: "Evaluated" in which way?

p.6 line 31: 50% of what? Of all pixel within a 100m x time from RS launch to 30min after RS end? Or 50% of pixel at a certain altitude?

p.6 line 34: Specify that the ice crystal is actually growing in the supersaturated layer and which microphysical growth processes could occur and specify in which way the ice crystal can "influence" the cloud.

p.7 line 12 -13: Why do you refer to seeding situations here when you describe non-seeding situations in line 10-11?

p.9 line 8-10: Describe the influence of varying ice crystal size more in detail: 57% of

possible seeding for 150micron ice particle size and only 37% for 50micron. . .

p.9 line 11: Please expand your discussion of Figure 5. In number indicate the number of cases considered for each month (February maybe has a much lower number of cases?).

p.10 line 2: Define acronym IN(P)

p.16 line 2: vertically-pointing cloud Doppler radar (instead of only "radar")

---

## Author Comment (AC1) · 27 Dec 2018

Review of "Classification of Arctic multilayer clouds using radiosoundings and radar data" by Vessel et al.

We thank the anonymous reviewer for his/her review and the detailed comments. We have revised the manuscript accordingly, including a revision of all sublimation calculations, updates of all figures, and major changes of the text. Our replies to your comments are given below in blue after the specific comment. Our page references refer to the corrected version of the paper.

Recommendation: Might be acceptable for publication after mandatory revision

This paper analyzes a year of data collected by a radar and radiosoundings at Ny-Alesund and attempts to determine the frequency of occurrence of multi-layer clouds, and in the case of multi-layer clouds whether the cloud layer underneath is seeded by the cloud above. The subject matter is timely because the Arctic is currently warming quicker than other parts of the planet, yet models have a difficult time accurately predicting the amount of warming. Better knowledge of the properties of arctic clouds, and on what controls them, is necessary in order to improve these predictions: any paper that hence contributes to our data base on the phases, heights and geometrical characteristics of clouds is beneficial. The paper is well written and the presented analysis easy to understand. Nevertheless, I fear that the paper as currently written is quite misleading. There are so many uncertainties and problems with the analysis (which, in their defense, the authors do a good job of identifying) that I fear the results that come out of the paper are not terribly useful. However, I think if the data were presented in an alternate way the study could be of potential use and hence I am recommending major revision rather than rejection.

Step 1 of their analysis uses the radiosonde data to identify the presence of multi-layer clouds. However, even though the probability of detection of the multi-layer clouds is 99% with this method, the false alarm rate of 58% "reveals that about half of the MLC detected by radiosounding is no MLC by radar." Thus, it seems that the paper should be reworded to emphasize that the use of the radiosonde data on its own does not reliably identify the occurrence of MLCs, but can be used in combination with radar data to give information on the presence of MLCs. The authors acknowledge the unreliability of the radiosonde data on their own to identify MLCs as they state "even if the layers above and below are supersaturated with respect to ice, the lack of suitable IN can prevent ice cloud formation." They also stated that "the results obtained by the radiosonde profiles disagree with actual MLC occurrence observed by the radar."

It is true that we find using only radiosonde data not being the best method to detect cloud layers. The use of radar instead would lead to much more reliable statistics about

visible multilayer clouds. For this we refer to Nomokonova et al. [2018]. However, our main idea was to investigate the possibility of seeding in connection with multilayer clouds. In order to do so, a radiosonde profile is essential to calculate the possibility of seeding (using the temperature and humidity profile of the radiosounding). Using primarily radar and in a second step the radiosonde does not solve the problem, since we show that seeding does hardly ever occur in between two in the radar visible cloud layers (cat. 8 in Fig. 8 in Vassel et al. [2018]). That means seeding itself can very poorly be differentiated from a cloud layer in the radar. Because of that we use primarily radiosonde data and radar data only as a further measure, even if this leads to high uncertainties. We have reworded some of the paragraphs in order to make clear that both radiosonde and radar is needed (p.10 l.16-17, p.15 l.16-17).

The second major problem with the analysis presented is the reliance on the chosen ice crystal size to calculate which of the upper layers of MLCs is seeding a lower layer. As stated by the authors "varying the initial ice crystal size has a large, non-linear impact on the distribution between seeding and non-seeding subsaturated layers." Further, their calculations substantially underestimate the variance that the size of the seeding ice crystal size might have. In several studies of in-situ measurements of mixed-phase clouds, the ice crystal sizes have been much larger than the 150 micrometer size assumed in the calculations here. Further, the calculations assume a hexagaonal plate which is not representative of the shapes of ice crystals in mixed-phase clouds. For example, Korolev et al. (1998) found that over 98% of ice crystals in mixed-phase arctic clouds had irregular shapes. Thus, the uncertainties will be much larger than those stated, and the stated uncertainties are already huge. And, the base size of 100 micrometers is probably much smaller than the size of particles that will be emanating from the upper layer.

This is a valid comment, and we have revised the calculations taking into account larger crystal sizes. The upper cloud of a MLC, from where the falling ice crystals origin, can either be a mixed-phase cloud or a cirrus cloud. In mixed-phase clouds Korolev et al. [1999] measured ice crystals with radii of about $r =400$ $\mu$m. We also refer to Fig. 5e in Mioche et al. [2016], where a radius of $r =400 - 500$ $\mu$m can occur in mixed-phase clouds. For cirrus clouds Krämer et al. [2009] showed that the radii of ice crystals range between $r =1 - 100$ $\mu$m. In order to account for both cloud types, we have redone our calculations for the ice crystal sizes $r =100, 200$ and $400$ $\mu$m. Our main focus is now on $r =400$ $\mu$m, assuming in most cases the upper cloud to be mixed-phase.

We agree with you that the ice crystal shape should not be treated as a sphere. We changed the calculation and the text accordingly (p.4 and p.6). We have selected the four ice crystal shapes hexagonal plate, rimed particle, stellar and irregular particle which are representative for mixed-phase clouds [Mioche et al., 2016]. For these particles we use the fall speed calculation shown in Fig. 1 and given by Mitchell [1996]. The main focus in the paper is on the hexagonal plate. The results for the other shapes is presented in the Appendix. We do not account for the lower limit of ice crystal size given by Mitchell [1996], since in our calculation we have to calculate the speed also for very small ice crystals due to sublimation. Note that this might lead to a small error of too fast falling

small ice crystals.

We also corrected the capacitance. We are now using the calculation of Westbrook et al. [2008]. The cases selected are listed in A1 of the paper, as well as the aspect ratios chosen by us.

In Fig. 2 we present the variation of the different ice crystal shapes on the result of classification step 2. As mentioned in the paper, on classification step 1 there is a small variation, but on classification step 2 there is almost no impact of the ice crystal shape on the result (p.10, l.4 and p.12, l.14-15).

[Figure]

Figure 1: Ice crystal fall speed in dependence of particle size [Mitchell, 1996]

[Figure]

Figure 2: Classification step 2 using different ice crystal shapes with $r = 400 \ \mu$m.

Another potential problem could be the lack of colocation between the radiosondes (which can drift large distances in the background wind) and the radar, which is again noted by the authors: "horizontal drift of the radiosonde away from the radar and inaccuracies due to time averaging of the radar data can explain contradictions between radiosounding and radar." Was any effort made to consider the advection of air parcels measured by the radiosondes so that the radar data at an appropriate time could be used in the analysis (provided that the air parcel was within the radar view volume at some time)?

In response to this comment, we have revised our calculations. In order to account for the advection, we calculate the wind speed in each layer. Using this information together with the distance between the radiosonde and the radar, we calculate the time the air parcel needs to drift from the radar to the position of the radiosonde. For this we do not consider the wind direction, but assume that the air parcel drifts the same direction as the radiosonde. As an example we show the results of the calculation for 3 November 2016 (Fig. 3). For the 3 November 2016 the average time over all heights is 12.94 min. For our statistics we have added the average time for each day to the time chosen for the evaluation of the radar data. For the 3 November 2016 the resulting radar time period is shown in Fig. 4. The results were changed only marginally by this correction.

[Figure]

Figure 3: Advection on 3 November 2016: a) windspeed in each height layer, b) estimated distance between radar and radiosonde, c) estimated time that the air parcel needs to drift from the radar to the position of the radiosonde.

[Figure]

Figure 4: a) Radiosonde data, b) radar time period corrected due to advection

MORE DETAILED COMMENTS:

Abstract: At first reading, I was confused that the 9% and 23% of the cases mentioned because it did not add to 100%. Perhaps mention that other cases (of the 8 categories) are included to avoid confusion.
Thanks for this comment, we added the following bracket to make it clearer: "Seeding cases are found often, in 23 % of the investigated days (100 % includes all days, also non-cloudy days)".

Page 1, Line 24: I find trying to differentiate between the terms multilayered clouds and multilayer clouds (MLC) very confusing! To me, that is the exact same word.
We agree with you that the use of the terms multilayered cloud/multilayer clouds create some confusion at the moment and are not used in a consistent way in literature. Verlinde et al. [2007, 2013] describe a multilayered cloud as a continuous cloud layer with a variable lidar signal inside this cloud layer. It refers to one cloud with different layers. We use the word multilayer clouds referring to separate clouds with clear visible interstice in between. In our case we refer to cloud layers in the atmospheric column, not layers within the cloud.

Page 3, line 6: Do you expect any diurnal cycle in the cloud properties that would mean that the derived statistics are not representative of the Arctic as a whole?
We did not investigate the diurnal cycle in the cloud properties. This is not possible since the radiosonde is most of the year only launched once per day (11 UTC). Since we do not expect any diurnal cycle, we consider the statistics as representative for the location as a whole. However we want to point out that there are differences in the weather at Ny-Ålesund compared to other locations in the Arctic due to the location in a fjord on the west coast of Svalbard compared to the typical sea ice influenced high Arctic. We consider Ny-Ålesund as an Arctic but not as a high Arctic location.

Page 3, line 10: I think data is plural, so it should be "data are" rather than "data is"
We have changed it at page 3, line 10. At other places either only radiosonde or only radar is used in connection with data. Then it is grammatically correct to use "radar data is".

Page 3, line 28: What statistical test was applied to show that the results did not change significantly?
No statistical test was applied. The use of the word significantly is wrong here. It is now changed to substantially.

Page 7, line 9: Why is only the lowermost 100 m considered?
We rephrased the sentence to make this more clear to "For the subsaturated layer in between only the lowermost 100 m are evaluated in order to address the question if the ice crystal survives so far. If the layer is thinner than 100 m only the available vertical thickness is considered." If there is no radar signal in this lowest part, then the ice crystal has not survived.

Page 10, line 11: Vali (200x?) has recommended that ice nucleating particles (INPs) rather than ice nuclei (IN) be used in order to standardize terminology. Recommend that you use INP rather than IN.
Thanks for the comment, we have changed it accordingly.

Page 15, line 1: The cloud layers can slope up and down frequently (in relatively short distances or times) and that can have a big impact on averaging. Was this taken into account?
The radar signal was averaged over a time of $\pm$ 30 min. Sometimes we have conditions like e.g. no cloud/high cloud at the start time (-30 min) and later (+30 min) a cloud reaching much lower (e.g. 3.7., 31.7., 18.8., 2.10., 14.2., 21.5.). In these cases the layer is almost half covered and half not covered and it is unclear if this should be counted as a cloud containing layer or not. Then averaging is the most consistent solution. Reducing the average time to $\pm$ 15 min does not improve the results, the Heidke skill scores are reduced in this case by 0.02 and 0.01.

Page 17, line 3: Perhaps I am not looking in the right place, but I cannot find the supplement being referred to in this statement.
You are completely right. This is now corrected.

**References**

A. Korolev, G. Isaac, and J. Hallett. Ice particle habits in arctic clouds. *Geophysical research letters*, 26(9):1299–1302, 1999.

M. Krämer, C. Schiller, A. Afchine, R. Bauer, I. Gensch, A. Mangold, S. Schlicht, N. Spelten, N. Sitnikov, S. Borrmann, et al. Ice supersaturations and cirrus cloud crystal numbers. *Atmospheric Chemistry and Physics*, 9(11):3505–3522, 2009.

G. Mioche, O. Jourdan, J. Delanoë, C. Gourbeyre, G. Febvre, R. Dupuy, M. Monier, F. Szczap, A. Schwarzenboeck, and J.-F. Gayet. Vertical distribution of microphysical properties of Arctic springtime low-level mixed-phase clouds over the Greenland and Norwegian seas. *Atmospheric Chemistry and Physics*, 17:12845–12869, 2016.

D. L. Mitchell. Use of mass-and area-dimensional power laws for determining precipitation particle terminal velocities. *Journal of the atmospheric sciences*, 53(12):1710–1723, 1996.

T. Nomokonova, K. Ebell, U. Löhnert, M. Maturilli, C. Ritter, and E. O'Connor. Statistics on clouds and their relation to thermodynamic conditions at ny-ålesund using ground-based sensor synergy. *Atmospheric Chemistry and Physics Discussions*, 2018:1–37, 2018. doi: 10.5194/acp-2018-1144. URL `https://www.atmos-chem-phys-discuss.net/acp-2018-1144/`.

M. Vassel, L. Ickes, M. Maturilli, and C. Hoose. Classification of Arctic multilayer clouds using radiosonde and radar data. *Atmospheric Chemistry and Physics Discussions*, 2018:1–22, 2018.

J. Verlinde, J. Y. Harrington, V. Yannuzzi, A. Avramov, S. Greenberg, S. Richardson, C. Bahrmann, G. McFarquhar, G. Zhang, N. Johnson, et al. The mixed-phase Arctic cloud experiment. *Bulletin of the American Meteorological Society*, 88(2):205–221, 2007.

J. Verlinde, M. P. Rambukkange, E. E. Clothiaux, G. M. McFarquhar, and E. W. Eloranta. Arctic multilayered, mixed-phase cloud processes revealed in millimeter-wave cloud radar Doppler spectra. *Journal of Geophysical Research: Atmospheres*, 118(23), 2013.

C. D. Westbrook, R. J. Hogan, and A. J. Illingworth. The capacitance of pristine ice crystals and aggregate snowflakes. *Journal of the Atmospheric Sciences*, 65(1):206–219, 2008.

---

## Author Comment (AC2) · 27 Dec 2018

We thank the anonymous reviewer for his/her review and the detailed comments. We have revised the manuscript accordingly, including a revision of all sublimation calculations, updates of all figures, and major changes of the text. Our replies to your comments are given below in blue after the specific comment. Our page references refer to the corrected version of the paper.

Summary of the manuscript

The study titled "Classification of Arctic multilayer clouds using radiosonde and radar data" by Maiken Vassel et al. describes an algorithm for the classification of multi-layer cloud occurrence for a one year dataset in Ny Alesund, Svalbard based on radiosonde and vertically-pointing cloud radar observations. The classification is two-fold: Firstly, only the conditions for cloud occurrence based on radiosonde humidity profiles consisting of two supersaturated layers separated by a subsaturated layer are analyzed. The fall distance of a hexagonal ice crystal of 100 micron size before complete evaporation in the subsaturated layer are estimated. The subsaturated layers are then classified into two categories. The first category is called "seeding", referring to layers with a vertical extent lower than the fall distance before complete ice crystal sublimation. The second category is called "non-seeding", referring to layers with a vertical extent higher than the fall distance before complete before complete ice crystal sublimation. These maximum possible occurrence frequencies for multi-layer cloud occurrence based on supersaturated layers as identified by radiosonde ascents are then verified by cloud radar reflectivity profiles obtained within 30min before radiosonde launch and 30min after the radiosonde has reached 10 km altitude. Multilayer mixed-phase cloud occurrence was found in 29% of the cases based on the combined radiosonde-cloud radar estimation. One of the main finding of the paper is that about 50% of the multilayer clouds estimated solely from radiosonde humidity profiles are not classified as such by the radar. - But the conclusion that radiosounding data is not sufficient for multi-layer cloud occurrence classification since not only humidity but also concentrations of ice nucleating particle (INP) and cloud condensation nuclei (CCN) are crucial is not made.

We added an explanation of the missing overlap of radiosonde and radar data on page 11 by referring to Spichtinger et al. (2002):" In the ice-supersaturated layers above and below missing cloud formation can be explained by the lack of aerosol as INP (Spichtinger et al., 2002)". Also in the conclusions we refer to this problem. Here we reworded the sentence to: "A high amount of supersaturated layers found in the radiosonde profiles does not coincide with observed cloud occurrence, probably due to lack of INP and thereby missing cloud formation" (page 17).

I would suggest the manuscript to be published after major revisions. The authors

should address the following points:

Major comments:

The literature review in the introduction should be extended. For example: p.1 line 20: Some more recent publications on Arctic mixed-phase cloud properties should also be included, for example Shupe 2011 (DOI: 10.1175/2010JAMC2468.1). In this paper. e.g., the occurrence of mixed-phase clouds at Arctic sites was found up to altitudes of 7-8 km.

Thanks for the comment. We have corrected this and added the suggested reference.

p.2 line 18: Please give a broader and more detailed introduction on seeder-feeder mechanism and why it is important to consider it in the Arctic. Also, be more specific in which way the ice crystals falling from the upper cloud "influence" the lower cloud. Since this is a central question of this study, it has to be properly introduced. The points made in various parts of the manuscript (especially Section 2.3) need to be more precise instead of using colloquial expressions like "surviving" ice crystals etc. (e.g., also: p.6 line 24-29; p. 10 line 1-16, p.11, line 29-30, p.16 lines 18-20, etc.: very colloquial language).

We have added a more detailed description about the possible outcomes of the seeder-feeder mechanism (page 2, line 17 onwards). Regarding the use of the term *survive*, we have now defined it on p.4, l.21 and use it further on. In some passages we have exchanged *surviving* with *not fully sublimated* (e.g. section 2.3).

p.6 line 24-29: We have moved the sentence about the detection limit to the section 2.1.:"The detection limit is -19.47 dBZ at 223 m, -57.31 dBZ at 423 m and -28.61 dBZ at 10 km".

p. 10 line 1-16: We have reworded the passage to: "For the seeding cases the cloud categories are shown in Fig. 8..." (p.10, l.21 - p.11, l.15).

p.11, line 29-30: We have changed it to: "A discontinuous radar signal only existing of small shreds of clouds is not counted as cloud and a continuous radar signal containing some small cloud free holes is counted as cloud."

p.16 lines 18-20: We have changed it to: "Following from our sublimation calculation we find that MLCs, which are clearly (visibly) separated in the radar, do not interact through seeding. However, we have to keep uncertainties like the radar detection limit in mind."

p.4 line 1-14: Please include a conceptual sketch of which kind of cloud layers you are considering indicating minimum depth of the layers, minimum vertical spacing between two supersaturated layers with a subsaturated layer between, temperature restrictions... otherwise it is really hard to follow.

On p.6 we have added a conceptual sketch of how we consider the radiosonde and radar data for potential MLCs.

p.4: It is mentioned that a simplified approach is used to determine the capacitance for

a hexagonal plate. (By how much) does the capacitance differ for different ice crystal shape assumptions (columns/dendrites/quasi-spherical spheres)? Also, why do you use a radius-volume relation of a sphere (p.5) if you consider hexagonal plates? As e.g. shown by Mitchell, JAS 1996, ice particle fall speed is a strong function of ice particle shape and density. Only assuming one particular ice crystal shape (hexagonal plates) is not sufficient for the fall distance estimation. Sublimation calculations should at least be repeated for two other ice particle shapes with very different fall speed characteristics such as columns and dendrites which would lead to very different fall distances. Additionally, please mention the influence of up- and downdrafts on ice particle fall velocity and thus fall distance.

The first reviewer also commented on the ice crystal shape. We agree with you that the ice crystal shape should not be treated as a sphere. We changed the calculation and the text accordingly (p.4 and p.6). We have selected the four ice crystal shapes hexagonal plate, rimed particle, stellar and irregular particle which are representative for mixed-phase clouds (Mioche et al., 2016). For these particles we use the fall speed calculation shown in Fig. 1 and given by Mitchell (1996). We selected the particles to be realistic for both mixed-phase clouds and cirrus clouds. Additionally we chose the four shapes where the falls speeds varies the most. The main focus in the paper is on the hexagonal plate. The results for the other shapes is presented in the Appendix. We do not account for the lower limit of ice crystal size given by Mitchell (1996), since in our calculation we have to calculate the speed also for very small ice crystals due to sublimation. Note that this might lead to a small error of too fast falling small ice crystals.

We also corrected the capacitance. We are now using the calculation of Westbrook et al. (2008). The cases selected are listed on p.19 in the paper, as well as the aspect ratios chosen by us. In the last column the calculated capacitances for $r = 400$ $\mu$m are shown. For the selected ice particle shapes the values range between $C = 2.00 \times 10^{-4}$ to $C = 3.88 \times 10^{-4}$. Earlier we used a capacitance of $C = 2.55 \times 10^{-4}$ for $r = 400$ $\mu$m.

We added a sentence pointing to the uncertainty due to the missing consideration of the up- and downdrafts.

In Fig. 2 we present the impact of the different ice crystal shapes on the result of classification step 2. As mentioned in the paper, for classification step 1 there is a small impact, but for classification step 2 there is almost no impact of the ice crystal shape on the result (p.10, l.4 and p.12, l.14-15).

p.6 line 24-29: Very colloquial language describing the radar reflectivity above the detection limit (p.6) - please rephrase. Please describe you averaging more in detail: I assume you refer to temporal averaging at each altitude? Please include a third Panel in Fig.2 showing the radar reflectivity sensitivity profile and the averaged reflectivity of the example case (for 50% data points within the considered time span).

We have reworded it in the text to "If more than 50 % of the selected radar data contain radar reflectivity factor data (coloured in Fig. 3b), then it is defined as cloud by the algorithm" (p.8,l.6 onwards). We consider a box with 100 m height (see dashed boxes

[Figure]

Figure 1: Ice crystal fall speed in dependence of particle size (Mitchell, 1996)

[Figure]

Figure 2: Classification step 2 using different ice crystal shapes with $r = 400$ $\mu$m.

in Fig. 2 in the paper) and within this box we do not average but evaluate if more than 50% of the pixels contain radar reflectivity data. In Fig. 3 we show the radar sensitivity and the averaged reflectivity for each layer for the 3 November 2016. The measured reflectivity is above the radar sensitivity limit. On p.3 we added the sentence: "The detection limit is -19.47 dBZ at 223 m, -57.31 dBZ at 423 m and -28.61 dBZ at 10 km, and the evaluated values are about these limits."

p.8: I haven't seen a definition for a cloud case – how long of a gap in time is needed to refer to a scene having two separate clouds at one altitude– one radar profile (30s?) or a few minutes? Or is this not considered at all and averaging over time around RS launch is done in a way that separate cloud occurrences at one height are averaged into "one"?

Two clouds at one altitude are not considered separately. The evaluation over time is

[Figure]

Figure 3: a) Radar reflectivity factor, b) Radar sensitivity: Minimum detectable radar reflectivity as a function of height. It includes the effect of ground clutter and gas attenuation but not liquid attenuation. The averaged reflectivity is shown for each layer.

done so that separate clouds at the same altitude are considered as one cloud within this timespan (±30 min).

p.10 lines 1-4: I do not fully agree with your conclusion that no cloud return in the radar data always means cloud-free conditions. There could be situations in which the sensitivity of the radar is not high enough (LWC and IWC too low). – I thus strongly suggest to also use available profiling lidar data (ceilometer) instead of only radar data to check for cloud occurrence in supersaturated conditions. Although the ceilometer suffers from full attenuation at sufficiently optically thick clouds, it will likely increase the number of detected cloud occurrences from ground-based remote sensing observations.

On p.11 we added the sentence: "Indeed, a very low liquid and ice water content could also result in a value below the radar sensitivity limit explaining these cases."

From 10.6.2016 onwards radar data was recorded in Ny-Ålesund and therefore we started our time period on this date. We would have liked to include micro-pulse lidar data as provided by `https://mplnet.gsfc.nasa.gov/data?v=V2&s=Ny_Alesund&t=20160616`, but this record ended at 16.06.2016. In Fig. 4b) we show the ceilometer attenuated backscatter coefficient for the 3 November 2016. It gives us the lowest cloud base height, but not any information on the additional cloud base heights above.

Moreover, in the Arctic frequently clouds occur at very low altitudes which might sometimes be below the lowest radar/lidar range gate. On page 16 (line 15-16) you mention

[Figure]

Figure 4: a) Radar reflectivity factor, b) Attenuated backscatter coefficient measured by Vaisala 910-nm CL51 lidar ceilometer in Ny-Ålesund

that a lidar would be useful. I strongly suggest making use of the existing ceilometer data (https://www.awipev.eu/awipev-observatories/cloud-cover/) in your study.

In the comment before we explained the problems when including the ceilometer. You are right, it is possible that we miss cloud layers existing only below 223 m. On p.3 we have added: "The detection height extend from 223 m until 10 km." and on p.11 we have added: "Additionally, the minimum detection height of 223 m might lead to some cases not considered." in order to point out that we might miss some cases. However, we have looked through this lowest layer in the relative humidity measurements and we do not think that there are many cases we miss.

Minor comments:

We have added all the following minor corrections.

Title: I suggest adding "in Svalbard" to narrow down the geographical range of the study. Also, since you are only considering $T < 0°C$, you can add "cold" clouds in the title.

The new title is now: "Classification of Arctic multilayer clouds using radiosonde and radar data in Svalbard". We have added "in Svalbard" but we did not add "cold". We do not find this specification necessary. There is almost no case of MLC that we miss due to the temperature restriction $< 0°C$.

p.4: For all variables in the equations the units should be included.

p.1 Line 10-13: It is unclear which kind of "deviations" you refer to – please specify

more in detail.

We have changed the sentence to the following: "Since there are various deviations between the relative humidity profiles and the radar images, e.g. due to horizontal wind drift and time restriction, an evaluation by manual visual inspection is additionally done for the non-seeding cases."

p.1 line 16: it should be "improve" instead of "improved"

p.1 line 16: hydrometeor shape and density (and thus terminal fall velocity) are of great importance, too.

p.1 line 21: You could add that the typical structure of stratiform mixed-phase cloud with supercooled liquid top layer and precipitating ice points to heterogeneous ice formation processes (add citation).

p.1 line 21: rather from the "remote sensing point of view"

p.1 line 22ff: "variable lidar signals inside a more or less continuous radar signal" sounds very imprecise – please rephrase and refer to "cloud profiles obtained with vertically-pointing instruments" or sth. similar

We have changed the sentence to:"From the remote sensing point of view, Verlinde et al. (2007, 2013) obtained cloud profiles with vertically-pointing instruments and described multilayered clouds as multiple distinct liquid layers within one vertical extensive cloud."

p.1 line 24: make it easier for the reader and put multilayered vs. multilayer in Italics or bold font

p.1 line 24: In which measurement is the interstice of multilayer clouds visible – in profiles of radar or lidar or both?

In both lidar and radar measurements the interstice of multilayer clouds could be visible. However, since the lidar is usually not able to penetrate the lower cloud layer, we focus here on radar measurements.

p.2 line 2: I suppose, you mean "at least" two clouds in different heights since there can be very low boundary layer clouds, midlevel clouds, and high-level clouds occurring simultaneously

p.2 line 20: Please modify the sentence since you look at one specific Arctic site (... occur at Ny Alesund not "the Arctic")

p.2 line 20: Why "thereby"?

We exchanged "Thereby we include..." with "We include..."

p.2 line 22: it should be "ground-based remote sensing" measurements

p.2 line 23: What do you mean by "easily accessible"?

Ground-based measurements are rare in the Arctic. Field campaigns are only limited to a short time period. Radiosondings have the great advantage, that they are conducted

all year around, each day and at multiple places all over the Arctic. In order to construct a classification, data availability over a full year is a great advantage. To compare various Arctic sites, the same type of measurement (radiosonde) is favourable.

p.2 line 25: be more precise in your wording: Instead of "radar" it should be profiling/zenith-pointing Doppler cloud radar

p.3 line 20: Please give a rough estimate of horizontal drift of the sondes based on their GPS tracking.

The first reviewer commented this as well. We present the same correction to both of you: In order to account for the horizontal drift, we calculate the wind speed in each layer. Using this information together with the distance between the radiosonde and the radar, we calculate the time at which the air parcel measured by the radiosonde was at the position of the radar. For this we do not consider the wind direction, but assume that the air parcel drifts the same direction as the radiosonde. As an example we show the results of the calculation for 3 November 2016 (Fig. 5). For the 3 November 2016 the average time over all heights is 12.94 min. For our statistics we add the average time for each day to the time chosen for the evaluation of the radar data. For the 3 November 2016 the resulting radar time period is shown in Fig. 6.

[Figure]

Figure 5: Advection on 3 November 2016: a) windspeed in each height layer, b)distance between radar and radiosonde, c) time the air parcel needs to drift from the radar to the position of the radiosonde.

p.3 line 23: Please also indicate the lowest radar range gate and mention that the cloud Doppler radar is zenith-pointing. You mention a vertical radar range gate resolution of 20m, is it really the same at all altitudes (i.e., all chirps) and was the RPG radar really operated in the same mode (with the same vertical and temporal resolution) over the entire year? 30s temporal resolution seems very low – please verify this temporal resolution... or are you using data already averaged to Cloudnet temporal resolution?

You are right, we use data already averaged to Cloudnet resolution. That is why our

[Figure]

Figure 6: a) Radiosonde data, b) radar time period corrected due to advection. At the time when the radiosonde reached the supersaturated layer 1 at 3.85 km the radiosonde is 3.70 km away from the radar due to horizontal wind drift.

data has a vertical resolution of 20 m at all heights and a temporal resolution of 30 s. The lowest range gate provided in our data set is 223 m. We have highlighted in the text that we use the averaged data. The original resolutions are different: The radar was operated in the high resolution mode. Here the vertical resolution varies from 4 m at 100 m height to 17 m at 10 km. The lowest range gate measured by the radar is 100 m. The temporal resolution is continuously 2.5 s.

p.3 line 25: Please indicate typical values of attenuation correction for the 94GHz radar at Ny Alesund.

Typical values of two-way radar attenuation due to atmospheric gases are between 0.09 dB at 223 m height and 1.20 dB at 10 km height (Fig. 7).

p.3 line 25: What do you mean by "at all frequencies"?

We have deleted "by all frequencies". It was formulated in this way as calibration convention in the data description, but for this specific radar there is only one frequency.

p.6 line 4: ice crystal size r refers to maximum dimension? Motivate the choices of ice crystal size of 50/100/150 microns by citing typical values found in Arctic clouds.

Already the first reviewer commented this. We present the same correction and answer to both of you: This is a valid comment, and we have revised the calculations taking into account larger crystal sizes. The upper cloud of a MLC, from where the falling ice crystals origin, can either be a mixed-phase cloud or a cirrus cloud. In mixed-phase

[Figure]

Figure 7: a) Radar reflectivity factor, b) Two-way radar attenuation due to atmospheric gases used for correcting Z.

clouds Korolev et al. (1999) measured ice crystals with radii of about $r = 400$ $\mu$m. We also refer to Fig. 5e in Mioche et al. (2016), where a radius of $r = 400$ - 500 $\mu$m can occur in mixed-phase clouds. For cirrus clouds Krämer et al. (2009) showed that the radii of ice crystals range between $r = 1$ - 100 $\mu$m. In order to account for both cloud types, we have redone our calculations for the ice crystal sizes $r = 100$, 200 and 400 $\mu$m. Our main focus is now on $r = 400$ $\mu$m, assuming in most cases the upper cloud to be mixed-phase. With $r$ we refer to the maximum dimension, $r = D_{\max}/2$.

p.6 line 4: It is unclear what you mean by "mean conditions": Mean over one hour after radiosonde launch?

By mean conditions we mean the average of the variable (pressure, temperature, humidity) over the height levels of the specific subsaturated layer.

p.6 line 5: "survive" is very colloquial, please replace or describe what you mean. Please change accordingly throughout the text.

We have now defined seeding on p.4, l.20 and use it therefore in the following text. In some passages we have exchanged *surviving* with *not fully sublimated* (e.g. p.6 l.11).

p.6 line 17: The way it is presented it sounds like as if the radar is used to test for cloud occurrence (above/between/below) in general and not only for cloud occurrence in general (Also see Table 1). (?)

We have corrected the sentence to: "The aim of adding radar data to the classification is to cross-check the super- and subsaturated layers in the radiosonde profiles with actual

cloud occurrence."

p.6 line 19-20: Clarify why you use 30min after the radiosonde reached 10km as end time and not e.g. simply a one hour time around RS launch (start 30min before and end 30min after launch)?

If we simply use a one hour time slot around the radiosonde launch start, we have cases (30.6.16) where in the height levels above 8 km the averaging of the radar data ends before the launched radiosonde reaches this height level. In order to avoid this we use + 30min after the radiosonde reached 10km as end time.

p.6 line 29-30: "Evaluated" in which way?

We have changed it to: "For the *subsaturated layer in between* only the lowermost 100 m are analysed in order to address the question if the ice crystal survives so far."

p.6 line 31: 50% of what? Of all pixel within a 100m x time from RS launch to 30min after RS end? Or 50% of pixel at a certain altitude?

We mean the 50% data points in the area of 100 m x time from RS launch (- 30min + advection time) to RS end (+ 30min + advection time).

p.6 line 34: Specify that the ice crystal is actually growing in the supersaturated layer and which microphysical growth processes could occur and specify in which way the ice crystal can "influence" the cloud.

We have changed it to: "The ice crystal begins to grow and can influence a cloud no matter at which height it is within the supersaturated layer." Further details on the influence of additional ice crystals on the cloud is mentioned when explaining the seeder-feeder effect (p.2,l.16-26).

p.7 line 12 -13: Why do you refer to seeding situations here when you describe non-seeding situations in line 10-11?

We have resorted and reformulated the sentences. We want to point out which two cloud categories we have chosen to be MLC cases (one non-seeding and one seeding case).

p.9 line 8-10: Describe the influence of varying ice crystal size more in detail: 57% of possible seeding for 150micron ice particle size and only 37% for 50micron...

We reworded it at p.10, l.6-l.12.

p.9 line 11: Please expand your discussion of Figure 5. In number indicate the number of cases considered for each month (February maybe has a much lower number of cases?).

We have added the number of cases (days) to Fig 5. You are right, during February there are less analysed days. This is due to the lack of radar data during this month. We reworded and expanded section 3.1.

p.10 line 2: Define acronym IN(P)

We already defined INP on page 2.

p.16 line 2: vertically-pointing cloud Doppler radar (instead of only "radar")

**References**

Korolev, A., Isaac, G., and Hallett, J.: Ice particle habits in Arctic clouds, Geophysical research letters, 26, 1299–1302, 1999.

Krämer, M., Schiller, C., Afchine, A., Bauer, R., Gensch, I., Mangold, A., Schlicht, S., Spelten, N., Sitnikov, N., Borrmann, S., et al.: Ice supersaturations and cirrus cloud crystal numbers, Atmospheric Chemistry and Physics, 9, 3505–3522, 2009.

Mioche, G., Jourdan, O., Delanoë, J., Gourbeyre, C., Febvre, G., Dupuy, R., Monier, M., Szczap, F., Schwarzenboeck, A., and Gayet, J.-F.: Vertical distribution of microphysical properties of Arctic springtime low-level mixed-phase clouds over the Greenland and Norwegian seas, Atmospheric Chemistry and Physics, 17, 12 845–12 869, 2016.

Mitchell, D. L.: Use of mass-and area-dimensional power laws for determining precipitation particle terminal velocities, Journal of the atmospheric sciences, 53, 1710–1723, 1996.

Spichtinger, P., Gierens, K., and Read, W.: The statistical distribution law of relative humidity in the global tropopause region, Meteorologische Zeitschrift, 11, 83–88, 2002.

Westbrook, C. D., Hogan, R. J., and Illingworth, A. J.: The capacitance of pristine ice crystals and aggregate snowflakes, Journal of the Atmospheric Sciences, 65, 206–219, 2008.

---

## Editor Decision (ED1)

**2nd review of Vassel et al., ACP 2018**

**Summary of the manuscript acp-2018-774**

The study titled "Classification of Arctic multilayer clouds using radiosonde and radar data in Svalbard" by Maiken Vassel et al. describes an algorithm for the classification of multi-layer cloud occurrence for a one year dataset in Ny Alesund, Svalbard based on radiosonde and vertically-pointing cloud radar observations. The classification is two-fold: Firstly, only the conditions for cloud occurrence based on radiosonde humidity profiles consisting of two supersaturated layers separated by a subsaturated layer are analyzed. The fall distances of a hexagonal ice crystal of 100/200/400 micron size before complete evaporation in the subsaturated layer are estimated. The subsaturated layers are then classified into two categories. The first category is called "seeding", referring to layers with a vertical extent lower than the fall distance before complete ice crystal sublimation – it was observed during 23% of the investigated days. The second category is called "non-seeding", referring to layers with a vertical extent higher than the fall distance before complete ice crystal sublimation. These maximum possible occurrence frequencies for multi-layer cloud occurrence based on supersaturated layers as identified by radiosonde ascents are then verified by cloud radar reflectivity profiles obtained within 30min before radiosonde launch and 30min after the radiosonde has reached 10 km altitude.
Multilayer mixed-phase cloud occurrence was found in 8-29% of the cases (depending on assumed ice crystal size, shape, and radiosonde humidity error) based on the combined radiosonde-cloud radar estimation..

**General Comments:**

The re-submitted version of the manuscript has improved with respect to the original submission by including more precise wording and extending the analysis. The authors addressed the comments made in the first review sufficiently. Specifically, sublimation calculations (of fall speed and ice crystal mass change with time) of ice crystals of varying sizes which are realistic for the considered clouds (radius of 100/200/400 microns) and their impact on seeding probability was included. The study was also extended by a sensitivity study on the influence of varying ice crystal shape (hexagonal plate, rimed column, sector plate, aggregate) in the Appendix. Moreover, a sensitivity study on how the classification results would change when considering a radiosonde humidity of +/- 5% was included in the Appendix.

The conclusion that radio sounding data alone is not sufficient for multi-layer cloud occurrence classification since not only liquid/ice saturation but also concentrations of ice nucleating particle (INP) and cloud condensation nuclei (CCN) are crucial is now made in the results- and conclusions sections but should also be mentioned in the abstract.

Also, please include a statement that no lidar Micro-Pulse Lidar (MPL) - which would have improved the cloud statistics in cases of clouds with low liquid water paths that are missed by the cloud radar - was available during your observation time period.

Even though the readability has improved, there is room for further improvement by shortening sentences or splitting them or simplifying the sentence structure.

I would suggest the manuscript to be published after minor revisions. The authors should address the following points:

**Minor comments**

p.8 Fig 4: Mention the assumed ice crystal shape used in the simulations for this plot.

Appendix:
p.19: Please refer to the included Table A1 in the main part of the manuscript. Table A1 should be extended by a terminal particle fall speed value for each assumed ice crystal shape.
p.20: Fig A2+A3: Regarding the radiosonde, mention that the +/-5% uncertainty is for the relative humidity.

---

## Author Response (AR2)

**Answer to 2nd review**

We thank the anonymous reviewer for his/her second review and the comments. We have revised the manuscript accordingly. We have corrected small things and simplified some sentences. Our replies to your comments are given below in blue after the specific comment. Our page references refer to the corrected version of the paper.

**2nd review of Vassel et al., ACP 2018**

**Summary of the manuscript acp-2018-774**

The study titled "Classification of Arctic multilayer clouds using radiosonde and radar data in Svalbard" by Maiken Vassel et al. describes an algorithm for the classification of multi-layer cloud occurrence for a one year dataset in Ny - Alesund, Svalbard based on radiosonde and vertically-pointing cloud radar observations. The classification is two-fold: Firstly, only the conditions for cloud occurrence based on radiosonde humidity profiles consisting of two supersaturated layers separated by a subsaturated layer are analyzed. The fall distances of a hexagonal ice crystal of 100/200/400 micron size before complete evaporation in the subsaturated layer are estimated. The subsaturated layers are then classified into two categories. The first category is called "seeding", referring to layers with a vertical extent lower than the fall distance before complete ice crystal sublimation – it was observed during 23% of the investigated days. The second category is called "non-seeding", referring to layers with a vertical extent higher than the fall distance before complete ice crystal sublimation. These maximum possible occurrence frequencies for multilayer cloud occurrence based on supersaturated layers as identified by radiosonde ascents are then verified by cloud radar reflectivity profiles obtained within 30min before radiosonde launch and 30min after the radiosonde has reached 10 km altitude.

Multilayer mixed-phase cloud occurrence was found in 8-29% of the cases (depending on assumed ice crystal size, shape, and radiosonde humidity error) based on the combined radiosonde-cloud radar estimation..

**General Comments:**

The re-submitted version of the manuscript has improved with respect to the original submission by including more precise wording and extending the analysis. The authors addressed the comments made in the first review sufficiently. Specifically, sublimation calculations (of fall speed and ice crystal mass change with time) of ice crystals of varying sizes which are realistic for the considered clouds (radius of 100/200/400 microns) and their impact on seeding probability was included. The study was also extended by a sensitivity study on the influence of varying ice crystal shape (hexagonal plate, rimed column, sector plate, aggregate) in the Appendix. Moreover, a sensitivity study on how the classification results would change when considering a radiosonde humidity of +/- 5% was included in the Appendix.

The conclusion that radio sounding data alone is not sufficient for multi-layer cloud

occurrence classification since not only liquid/ice saturation but also concentrations of ice nucleating particle (INP) and cloud condensation nuclei (CCN) are crucial is now made in the results- and conclusions sections but should also be mentioned in the abstract.

On p.1 l.11 we changed the sentence: " There are various deviations between the relative humidity profiles and the radar images, e.g. due to the lack of ice nucleating particles (INP) and cloud condensation nuclei (CCN) but also due to horizontal wind drift and time restriction. In order to account for some of these deviations an evaluation by manual visual inspection is done for the non-seeding cases."

Also, please include a statement that no lidar Micro-Pulse Lidar (MPL) - which would have improved the cloud statistics in cases of clouds with low liquid water paths that are missed by the cloud radar - was available during your observation time period.

We added it on p. 11, l.13 and on p.17, l.17.

Even though the readability has improved, there is room for further improvement by shortening sentences or splitting them or simplifying the sentence structure.

We have simplified some sentences.

I would suggest the manuscript to be published after minor revisions. The authors should address the following points:

**Minor comments**

p.8 Fig 4: Mention the assumed ice crystal shape used in the simulations for this plot.

We changed the sentence to: "The evaluated ice crystals are hexagonal plates with the initial sizes $r = 100$ $\mu$m, $200$ $\mu$m and $400$ $\mu$m."

**Appendix:**

p.19: Please refer to the included Table A1 in the main part of the manuscript. Table A1 should be extended by a terminal particle fall speed value for each assumed ice crystal shape.

On p. 10, l. 11 we added the reference: "The impact of the different ice crystal shapes (see Table A1) is shown in Fig. A1." In Table A1 we added a typical value for the fall speed.

p.20: Fig A2+A3: Regarding the radiosonde, mention that the +/-5% uncertainty is for the relative humidity.

We have changed it.

**Classification of Arctic multilayer clouds using radiosonde and radar data in Svalbard**

Maiken Vassel[1], Luisa Ickes[2], Marion Maturilli[3], and Corinna Hoose[1]

[1]Institute of Meteorology and Climate Research, Karlsruhe Institute of Technology, Karlsruhe, Germany
[2]Department of Meteorology, Stockholm University, Stockholm, Sweden
[3]Alfred Wegener Institute for Polar and Marine Research, Potsdam, Germany

**Correspondence:** Maiken Vassel (maiken.vassel@alumni.kit.edu), Luisa Ickes (luisa.ickes@misu.su.se)

**Abstract.** Multilayer clouds (MLC) occur more often in the Arctic than globally. In this study we present the results of a detection algorithm applied to  radiosonde and radar data from an one-year time period in Ny-Ålesund, Svalbard. Multilayer cloud occurrence is found on 29 % of the investigated days. These multilayer cloud cases are further analysed regarding the possibility of ice crystal seeding, meaning that an ice crystal can survive sublimation in a subsaturated layer between two cloud layers when falling through this layer. For this we analyse profiles of relative humidity with respect to ice to identify super- and subsaturated air layers. Then the sublimation of an ice crystal of an assumed initial size of $r = 400\,\mu m$ on its way through the subsaturated layer is calculated. If the ice crystal still exists when reaching a lower supersaturated layer, ice crystal seeding can potentially take place. Seeding cases are found often, in 23 % of the investigated days (100 % includes all days, also non-cloudy days). The identification of seeding cases is limited by the radar signal inside the subsaturated layer. Clearly separated multilayer clouds, defined by a clear interstice in the radar image, do not interact through seeding (9 % of the investigated days).  There are various deviations between the relative humidity profiles and the radar images, e.g. due to the lack of ice nucleating particles (INP) and cloud condensation nuclei (CCN). Additionally, horizontal wind drift of the radiosonde and time restriction when comparing radiosonde and radar data cause further deviations. In order to account for some of these deviations, an evaluation by manual visual inspection is  done for the non-seeding cases.

**1 Introduction**

Clouds radiate downwards in the long-wave part of the spectrum and thereby warm the surface in the Arctic during most of the year (Shupe and Intrieri, 2004). However, the correct representation of cloud fraction, cloud water content and its phase, particle size, shape, density and cloud temperature is difficult but essential to improve weather forecasting (Barrett et al., 2017a, b). Therefore clouds are still a major contributor to uncertainty in both weather and climate prediction.

In the recent years, an emphasis of research has been on Arctic mixed-phase clouds (Andronache, 2018; Morrison et al., 2012; Loewe et al., 2017). These clouds occur frequently in the Arctic, at all heights up to $8\,km$, and exist in the temperature range between $-34\,°C$ to $0\,°C$ (Shupe, 2011; Intrieri et al., 2002). They often consist of a supercooled liquid layer at cloud top  with precipitating ice particles below  and this points to heterogeneous ice formation (Whale, 2018).

Verlinde et al. (2007, 2013) described *multilayered* clouds as multiple distinct liquid layers within one vertical extensive cloud. They obtained cloud profiles with vertically-pointing remote sensing instruments. In contrast to *multilayered* clouds, *multilayer* clouds (MLCs) are described as two separate clouds with a clear visible interstice in between (Tsay and Jayaweera, 1984; Intrieri et al., 2002; Khvorostyanov et al., 2001; Fleishauer et al., 2002; Liu et al., 2012). The coexistence of these at least two clouds in different heights can be explained by horizontally inhomogeneous advection (Luo et al., 2008). In the Arctic these clouds are often a boundary layer cloud and a higher mixed-phase or cirrus cloud. When large-scale meridional transport brings warm moist air into the Arctic, temperature and humidity inversions occur frequently (Nygård et al., 2014). Reaching supersaturation and in the presence of sufficient cloud condensation nuclei (CCN) and ice nucleating particles (INPs), the horizontal advection can result in cloud formation at multiple heights (Curry and Herman, 1985).

Christensen et al. (2013) analysed radar and lidar data collected by the satellites CloudSat (millimetre wavelength cloud profiling radar) and CALIPSO (Cloud–Aerosol Lidar and Infrared Pathfinder Satellite Observations) to investigate the occurence of MLCs. They found, excluding the Arctic, the global average occurrence of MLCs to be 11 % of the data. For the Arctic, Liu et al. (2012) analysed similar satellite data of CloudSat and CALIPSO and found Arctic MLCs to occur between 17-25 % of the investigated time. The contribution of the MLCs to the seasonal variation of Arctic cloud coverage is only very weak. Cloud detection by satellites is challenging in the Arctic. A poor thermal and visible contrast between clouds and the underlying surface of snow and ice and small radiative fluxes from the cold polar atmosphere are only some of the uncertainties (Liu et al., 2012). Therefore and since the minimum considered layer thickness for separation was 960 m, Liu et al. (2012) assumed their estimated MLC occurence most likely to be underestimated.

Microphysical interaction between MLC layers can happen through the seeder-feeder mechanism (Fleishauer et al., 2002; Avramov and Harrington, 2010; Hobbs and Rangno, 1998; Houze Jr, 1993). This means that falling ice crystals from the upper cloud enrich the lower cloud by additional ice crystals. These ice crystals have then an influence on the evolution of the lower cloud's phase (e.g. glaciation). Inside the lower cloud vapour deposit onto the ice crystals causing ice crystal growth. 
[revised manuscript text omitted]
  the cloud is within this supersaturated layer.  Therefore, for the *supersaturated layer below* the algorithm  starts from the top  and searches for any layer of 100 m containing more than 50 % radar reflectivity factor data. If no layer of 100 m is containing more than 50 % radar reflectivity

10 factor data, then the evaluated vertical thickness is decreased until 20 m  at the lower boundary. If still no layer is containing more than 50 % radar reflectivity factor data, then it is considered that no cloud is present in this  *supersaturated layer below* (*no cloud*). In the example of 3 November 2016 (Fig. 3)  for the subsaturated layer 1 the *supersaturated layer above* is cloud containing (*cloud above*), the subsaturated layer 1 is not cloud containing (*no cloud in between*) and the *supersaturated*

[revised manuscript text omitted]

*Author contributions.* Corinna Hoose conceived the project, Maiken Vassel, Luisa Ickes and Corinna Hoose designed the analysis, Marion Maturilli provided the radiosonde data, and Maiken Vassel carried out the analysis. All authors contributed to the interpretation of the results. Maiken Vassel wrote the paper with input from all co-authors.

*Competing interests.* The authors confirm that they have no conflict of interest.

5   *Acknowledgements.*  This project has received funding from the European Research Council (ERC) under the European Union's Horizon 2020 research and innovation programme under grant agreement No 714062 (ERC Starting Grant "C2Phase"). Furthermore, we gratefully acknowledge scientific 
[revised manuscript text omitted]